# Extreme temperatures compromise male and female fertility in a large desert bird

Mads F. Schou [1✉], Maud Bonato [2], Anel Engelbrecht [3], Zanell Brand[3], Erik I. Svensson [1], Julian Melgar [1], Pfunzo T. Muvhali [2], Schalk W. P. Cloete [2,3] & Charlie K. Cornwallis[1]

Temperature has a crucial influence on the places where species can survive and reproduce. Past research has primarily focused on survival, making it unclear if temperature fluctuations constrain reproductive success, and if so whether populations harbour the potential to respond to climatic shifts. Here, using two decades of data from a large experimental breeding programme of the iconic ostrich (*Struthio camelus*) in South Africa, we show that the number of eggs females laid and the number of sperm males produced were highly sensitive to natural temperature extremes (ranging from −5 °C to 45 °C). This resulted in reductions in reproductive success of up to 44% with 5 °C deviations from their thermal optimum. In contrast, gamete quality was largely unaffected by temperature. Extreme temperatures also did not expose trade-offs between gametic traits. Instead, some females appeared to invest more in reproducing at high temperatures, which may facilitate responses to climate change. These results show that the robustness of fertility to temperature fluctuations, and not just temperature increases, is a critical aspect of species persistence in regions predicted to undergo the greatest change in climate volatility.

[1] Department of Biology, Lund University, Lund, Sweden. [2] Department of Animal Sciences, University of Stellenbosch, Matieland, South Africa. [3] Directorate Animal Sciences, Western Cape Department of Agriculture, Elsenburg, South Africa. ✉email: mads.schou@biol.lu.se

The range of temperatures that organisms can tolerate has a crucial influence on their distributions across space and time[1–3]. Our current understanding of thermal tolerance largely comes from studies examining how high temperatures affect survival[4–7]. However, it has recently been argued that because reproductive failure often occurs well before death, temperature effects on fertility (thermal fertility limits) may be more important in determining species responses to environmental change[8–12]. Characterizing how natural temperature fluctuations affect investment in fertility traits, such as the number and viability of eggs and sperm, and the impact this has on reproductive success is therefore crucially important, especially as climatic variation is expected to increase globally[13,14]. Do extreme temperatures have damaging effects on different fertility traits and if so, is there the potential for selection to increase resilience to changing climates?

Responses to selection for coping with more extreme and unpredictable temperatures relies on individuals varying in their thermal resilience[15]. One factor that can influence individual variation in thermal resilience is how reproductive and somatic investment are managed under thermal stress. For example, temperature extremes may lead to high physiological demand to protect essential organismal functions that reduce investment in reproduction[1,16,17]. Reduced reproductive investment can in turn generate trade-offs between different fertility traits that limit responses to selection for increased resilience to temperature change. However, whether temperature extremes expose such reproductive trade-offs, and the extent to which individuals vary in their prioritization of investment across different fertility traits, is unclear.

Research on the effects of natural temperature variation on reproduction in non-domesticated endotherms has primarily been on temperate species[18–33]. However, temperature unpredictability is greatest in tropical and sub-tropical regions and climate modelling shows this will increase in the future[13,34]. The reproductive performance of species living in such regions may also be particularly sensitive to the effects of climatic fluctuations, as they often have prolonged breeding seasons that increase

their risk of exposure to shifts in environmental conditions. Furthermore, because temperate species typically have short breeding seasons, timed to the seasonal appearance of food (phenology), there has been a focus on whether advancing spring temperatures reduce breeding success through phenological mismatches[18–28,35,36]. Consequently, more information is needed on the effects of ecologically relevant temperatures on investment in the traits directly related to fertility, such as the production and viability of eggs and sperm.

Here we examine how temperature fluctuations over a 20-year period affect multiple fertility traits in the world's largest bird, the ostrich (*Struthio camelus*), which reproduce throughout the year in tropical and sub-tropical regions (Fig. 1)[37–39]. Individually marked birds (*n* = 1299, Supplementary Table 1) were studied in the Klein Karoo region of South Africa where temperatures during the reproductive cycle ranged from −5 to 45 °C. Data on the fertility of females and males was obtained by collecting eggs daily from captive pairs, and by collecting natural ejaculates from captive solitary males. All pairs and solitary males used for sperm collection were kept in separate fenced enclosures of natural Karoo scrub exposed to natural weather conditions (Fig. 1a). Data were matched with onsite temperature records to investigate: (1) how thermal fluctuations shape investment in gametic traits (number of eggs and sperm, egg mass and sperm viability) and reproductive success (hatching success and offspring numbers), (2) individual variation in the resilience of fertility to temperature change, and (3) whether extreme temperatures cause trade-offs in investment across gametic traits.

## Results

**Is fertility compromised by hot and cold temperatures?** The number of eggs females laid and the number of sperm males ejaculated were significantly reduced by both increases and decreases in ambient temperature (Fig. 2a, b). The effects of temperature were not immediate, but resulted from a critical thermal window 2–4 days before laying and ejaculation (Supplementary Figs. 1 and 2; see the subsection "Time lag effects of

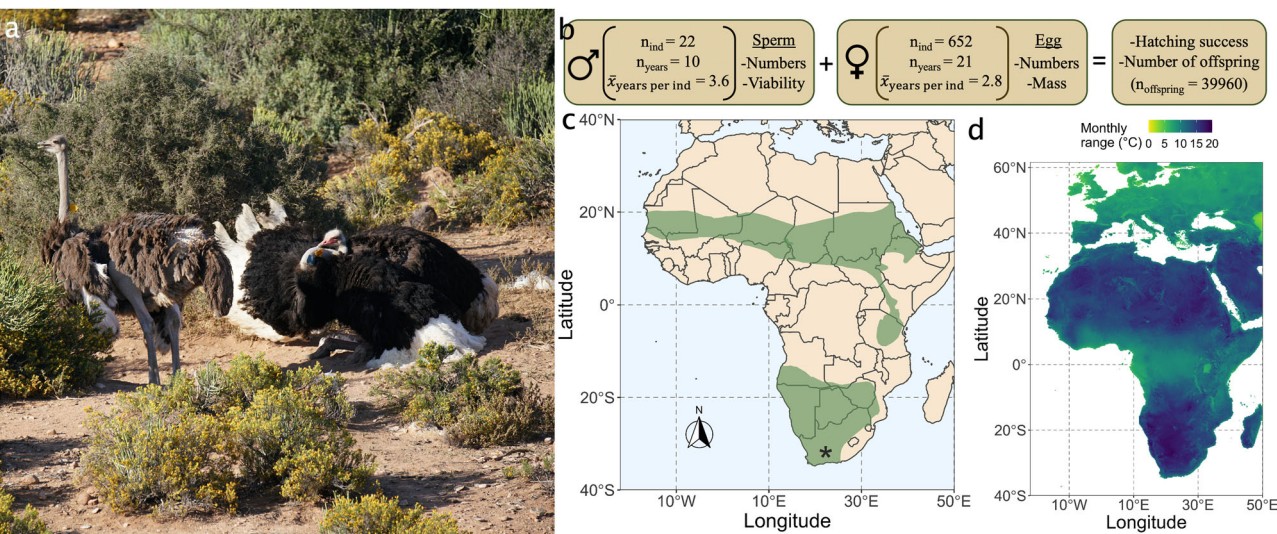

**Fig. 1 Ostriches (*Struthio camelus*) cope with large thermal fluctuations in their native habitat, reproducing successfully across Africa from the Western Cape to the deserts of Southern and Northern Africa. a** Courtship by a male ostrich (right) towards a female (left) in one of the enclosures (*n* = 197) at the study site used to keep a single breeding pair (photo: CKC). **b** Data structure of fertility traits obtained from 1998 to 2018 at the study site of Oudtshoorn Research Farm in the arid Klein Karoo region of South Africa. Sperm viability data was not available for all of the solitary males where measures of sperm numbers were obtained (sperm viability: $n_{ind} = 18$, $n_{years} = 7$, $\bar{x}_{years per ind} = 2.7$). See also Supplementary Table 1 for detailed overview of sample sizes. **c** Geographic range (green) of the ostrich[93] with the study site marked by an asterisk. **d** Monthly temperature range was calculated by estimating the range of temperatures of each month and then calculating the mean of this across all months[94].

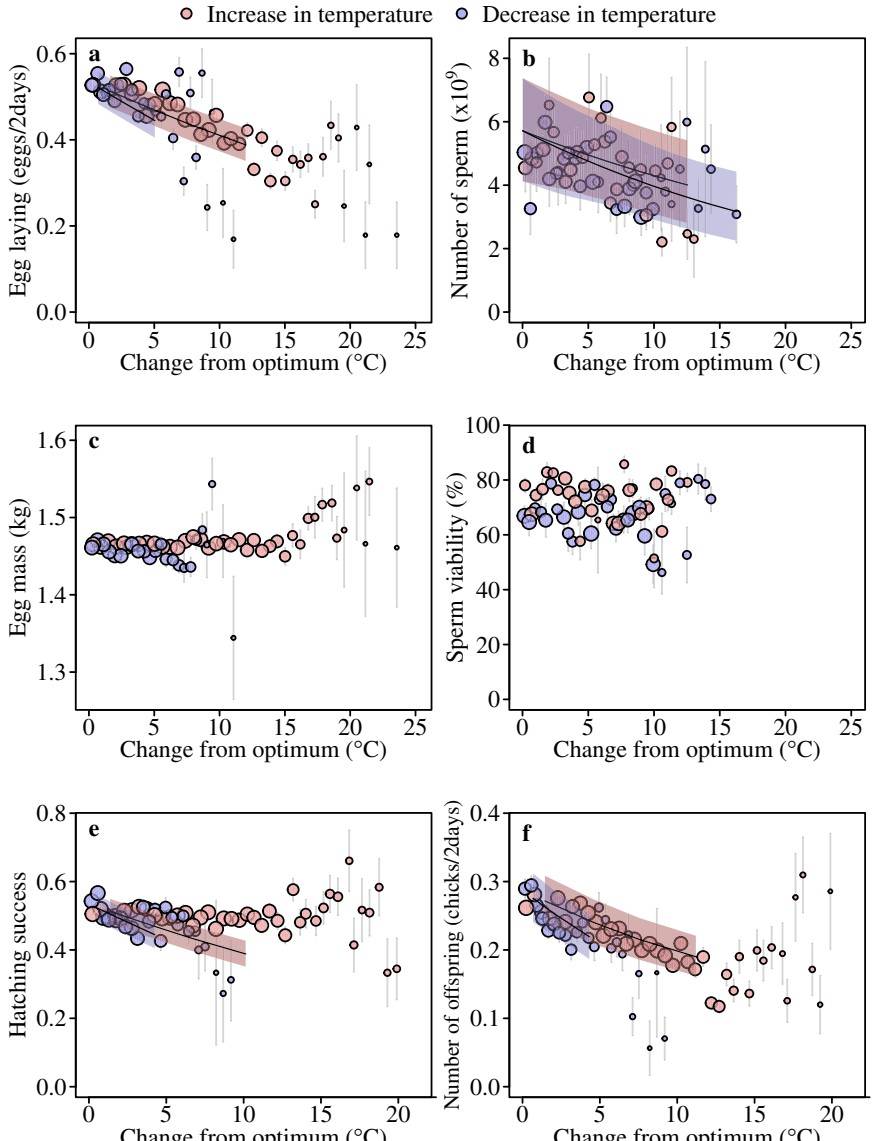

**Fig. 2 Temperature extremes compromise male (*n* = 22) and female (*n* = 652) fertility.** Female egg laying rate (**a**) and number of sperm ejaculated by males (**b**) were both highly sensitive to increases and decreases in temperature. Female (**c**: egg mass) and male (**d**: sperm viability) gamete quality were generally more resistant to temperature change. Hatching success (**e**), which is influenced by the egg mass[41,42], sperm numbers and sperm viability, was also less affected by temperature change. The number of offspring (**f**) is a product of hatching success and rates of egg laying and was influenced by changes in temperature that occured during egg laying. Ostrich females can only lay an egg every other day and we therefore used number of eggs or chicks per number of two-day intervals (eggs/2 days or chicks/2 days) (see the subsection "Time lag effects of temperature on gametes" in "Methods" section). The range of temperatures that sperm traits were measured at differed from the other traits, because it was not possible to collect sperm across all years (Supplementary Table 1). Fitted lines and 95% credible intervals (shaded area) from the primary set of models are shown for traits significantly affected by temperature (Supplementary Tables 2–7). For binomial models the fitted lines span the modelled binned temperature classes making them robust to outliers. Points are averages with standard errors binned according to the temperature variable. Point size illustrates relative number of observations. Source data are provided as a Source Data file.

temperature on gametes" in "Methods" section). During this critical thermal window, egg laying rate peaked at 20 °C (Supplementary Fig. 3), dropping by 15% and 18% when temperatures increased and decreased by 5 °C, respectively (Fig. 2a; Table 1, Supplementary Table 2). Similar reductions were seen in the number of sperm males ejaculated (19% with 5 °C increases and decreases from the optimum; Fig. 2b; Table 1, Supplementary Table 3), but the thermal optimum appeared to be slightly higher than for egg laying, peaking at ~26 °C (Supplementary Fig. 4). While this may indicate there is the potential for conflict over the thermal optima of males and females, this dataset was not designed to test this (see the subsection "thermal stress index" in

"Methods" section). It is also likely that both 20 and 26 °C are within the thermal neutral zone (TNZ), which although not explicitly known for ostriches, spans from 10–15 to 30 °C in the closest relative, the emu (*Dromaius novaehollandiae*)[40].

Fluctuations in temperature had much less of an effect on gamete viability than on the number of gametes. The mass of eggs females produced only decreased by 0.7% when temperatures fell from 20 to 15 °C and were unaffected by increases in temperature (Fig. 2c; Table 1; Supplementary Table 4). Similarly, the viability of sperm (viable sperm: normal morphology, intact membrane and eosin impermeable) males produced was robust to temperature fluctuations, with no consistent change with increases or

**Table 1 Individual variation in the resilience of fertility to temperature change.**

| Trait | Fixed effects (CI) | | | Repeatability (CI) | | | PSlopeVar (CI) |
|---|---|---|---|---|---|---|---|
| | Intercept | Slopes | | Intercept | Slopes | | |
| | | $T_{heat\ stress}$ | $T_{cold\ stress}$ | | $T_{heat\ stress}$ | $T_{cold\ stress}$ | |
| Egg laying | 0.32 (0.18,0.49) | −1.68 (−1.90,−1.50)*** | −2.10 (−2.43,−1.80)*** | 0.27 (0.21,0.33) | 0.24 (0.17,0.32) | 0.18 (0.06,0.34) | 0.16 (0.14,0.18) |
| Egg mass | 1.43 (1.42,1.45) | 0 (−0.01,0.02) | −0.05 (−0.07,−0.03)*** | 0.62 (0.59,0.65) | 0.52 (0.48,0.57) | 0.47 (0.41,0.53) | 0.03 (0.03,0.04) |
| Number of sperm | 10.46 (10.16,10.79) | −0.67 (−1.30,−0.02)* | −0.67 (−1.13,−0.23)** | 0.23 (0.13,0.39) | 0.47 (0.13,0.84) | 0.57 ((0.24,0.79) | 0.07 (0.04,0.13) |
| Sperm viability | −1.91 (−2.19,−1.62) | 0.13 (−0.24,0.42) | −0.21 (−0.55,0.11) | 0.54 (0.35,0.74) | 0.61 (0.38,0.81) | 0.60 (0.36,0.81) | 0.1 (0.05,0.17) |

We quantified the differences between individuals relative to within and between individual variation (repeatability) for fertility at intermediate temperatures (intercept) and for the change in fertility with increasing and decreasing temperatures (slopes). Estimates and credible intervals (CI) were extracted from the second set of MCMCglmm models including individual by year slopes. See Supplementary Tables 8–11 for model details including estimates of repeatability on the expected scale and variance of fixed effects. PSlopeVar = ratio of the slope variance to the total phenotypic variance.
*pMCMC < 0.05, **pMCMC < 0.01, ***pMCMC < 0.001.

decreases in temperature (Fig. 2d; Table 1; Supplementary Table 5).

**Do changes in fertility traits matter for reproductive success?** The effect of temperature on reproductive success (number of offspring) is a product of changes in egg laying rates and the probability that eggs hatch. Hatching success is in turn influenced by the fertilizing ability of males, which depends on the numbers and viability of sperm inseminated, and egg viability, which is linked to egg mass[41,42]. The potential effects of ambient temperatures during incubation on hatching success were removed by artificially incubating eggs using an on-site hatchery. Hatching success was significantly affected by the temperature birds experienced prior to laying: hatching success was reduced by 4–7% with 5 °C increases and decreases from 20 °C (Fig. 2e; $T_{heat\ stress}$ (credible interval, CI) = −0.26 (−0.43, −0.09), pMCMC = 0.002; $T_{cold\ stress}$ (CI) = −0.57 (−0.98, −0.01), pMCMC = 0.028; Supplementary Table 6). Combined with changes in egg laying rates, this resulted in the total number of offspring decreasing by 28% with 5 °C increases, and 44% with 5 °C decreases in temperature from 20 °C (Fig. 2f; $T_{cold\ stress}$ (CI) = −2.10 (−2.57, −1.60), pMCMC = 0.001; $T_{heat\ stress}$ (CI) = −1.42 (−1.61, −1.21), pMCMC = 0.001; Supplementary Table 7). Reproductive success can also be reduced if individuals die from temperature-related stress during the breeding season, but during the 21 years of experimental breeding only six adult deaths (0.5%) related to overheating were recorded. These results suggest that the negative effects of temperature fluctuations on reproductive success arise through the cumulative, detrimental effects on egg and sperm production under both low and high temperatures. It is also worth noting that these effects may be even more pronounced in wild populations where access to food and water is likely to be more restricted.

**Do individuals vary in how resilient their fertility is to temperature change?** There was substantial variation among females in how resilient their laying rates were to temperature change. Differences between individual females explained 24% of variation in the rate of decline in egg laying when temperatures increased, and 18% of variation when temperatures decreased (Table 1). Similarly, some males were much more resilient to temperature change than others, as indicated by the number of sperm they ejaculated (Table 1). When temperatures increased, 47% of variation in the decline in sperm numbers was explained by differences between males, and 57% when temperature decreased. We examined the robustness of these results using character state models where values of a trait are correlated between different temperature categories (cold (<17.7 °C), hot (>28.7 °C) and benign): correlations lower than one indicate variation between individuals in their response to temperature change[43]. These analyses confirmed that there were substantial differences among males and females in their responses to temperature change (Supplementary Tables 12 and 13).

Females were extremely consistent in their egg mass, which was relatively unaffected by temperature change (PSlopeVar: 0.03, Table 1). While average egg mass ranged from 1.41 to 1.68 kg among females, the most extreme change in egg mass of a female from 20 to 25 °C was an increase of just 0.015 kg. Despite this, a relatively large proportion of the variation in egg mass change was explained by differences between females, around 50%. Such consistent differences among females is in accordance with research on other bird species where egg mass is variable in populations, but highly consistent within individuals[44]. For males, the pattern was similar with around 60% of variation in the change in sperm viability with temperature being explained

by differences between males (Table 1). That said, character state models showed only a weak correlation between measures of sperm viability at benign versus cold and hot temperatures, suggesting that data from extreme temperatures may inflate the estimation of between individual differences (Supplementary Table 15). Taken together, these results show that when temperatures increase and decrease, individual females and males vary substantially in the number and viability of eggs and sperm they produce. The efficacy of selection to promote thermal tolerance is therefore unlikely to be limited by a lack of variation between individuals.

**Is the resilience of fertility to temperature change compromised by trade-offs between traits?** When individuals are exposed to temperature extremes, simultaneous investment in multiple traits may not be possible. The resulting trade-offs can take two forms. First, negative correlations between fertility traits may occur at extreme temperatures because physiological stress limits the resources individuals have to invest across reproductive traits. Second, there may be negative correlations in the degree of change across traits (thermal resilience) rather than absolute trait values. For example, investment in the maintenance of one trait may come at the expense of maintenance of other traits.

We found no evidence of any negative correlations between any fertility traits within or among individuals at any temperature (Fig. 3; Supplementary Table 18). This shows that the number of eggs females produce and the number of sperm males ejaculate is not traded-off against egg mass or sperm viability in either hot or cold periods. Instead, correlations between traits within females were generally significantly positive, indicating that investment in the number and mass of eggs are up and down regulated together (Fig. 3; Supplementary Table 18). Furthermore, among individuals there was a significant positive relationship between change in egg laying rates and change in egg mass as temperatures increased (Fig. 3). This is contrary to the idea that temperature stress induces trade-offs between fertility traits. Instead, this suggests that some females respond to higher temperatures by

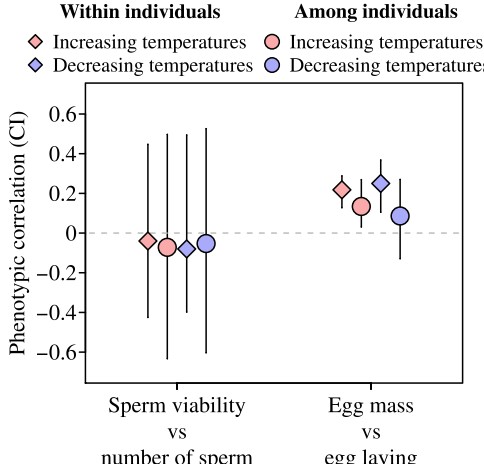

**Fig. 3 Correlated changes in the number and quality of gametes as temperatures increased and decreased.** The number of eggs and sperm females ($n = 652$) and males ($n = 18$) produced was not traded-off against egg mass and sperm viability as temperatures changed (see also Supplementary Table 18). This was consistent within and among individuals. Changes in egg-laying rates were positively correlated to egg mass as temperatures increased both within and among females (credible interval (CI) of phenotypic correlation excluded zero). Source data are provided as a Source Data file.

producing more eggs that are also heavier, compared to other females.

## Discussion

It has been argued that to understand how species are affected by environmental change, it is crucial to broaden the current focus on lethal limits to include thermal fertility limits[9]. Our results provide support for this proposition, as only six adults (0.5%) died from thermal stress, whereas there were dramatic reductions of 28–44% in reproductive success with 5 °C deviations from their thermal optimum. Although increased climatic change has brought into focus the effect of rising temperatures on survival and population persistence[34], our results show that cooler, as well as hotter, temperatures may pose a challenge for species.

Much of the classical life-history research on birds has focused on the seasonal appearance of food as a factor limiting breeding success[45,46], whereas the direct effects of temperature on reproduction have remained more unclear (but see Hurley et al.[11]). In ectotherms, extreme temperatures have been shown to reduce both the number and the quality of gametes individuals produce[9,47,48] and similar effects have been found in domestic chickens, domestic mammals and laboratory mice[49–52]. Such concordant effects of heat stress on different gametic traits suggests that high temperatures may lead to a general degradation of reproductive function. While our results show that heat and cold stress compromise reproductive success, this was not because of consistent detrimental effects across all traits, but rather specific responses of traits to temperature change: Sperm viability and egg mass did not decline even under the most extreme thermal stress, whereas the number of gametes individuals produced was highly sensitive to temperature change.

One potential reason for why the numbers and quality of gametes differ in their response to temperature change is that they are under different mechanistic control. Reductions in sperm and oocyte production caused by heat stress have been shown in mammals to occur due to decreases in testosterone in males and changes in luteinizing hormone in females[50,51]. General physiological changes due to temperature stress may therefore reduce rates of gametogenesis[9]. In contrast, changes in sperm and follicular function have previously been linked to processes, such as DNA damage[9,49–51], and may be somewhat shielded from physiological stress by the follicle/testes–blood barriers[53]. Alternatively, the limited effects of temperature on sperm viability and egg mass may be due to reduced sensitivity to physiological stress, consistent with early life-history models[54], or other measures of gametic performance may be required to detect the effects of temperature on gamete quality. For example, the biochemical composition of eggs can vary independently of egg mass and can influence offspring fitness[55,56]. The differences in the response of gametic traits to temperature change highlights the importance of understanding reproductive mechanisms when predicting outcomes of environmental change, and has important implications for how thermal fertility limits are studied.

The evolution of increased thermal tolerance is key to the persistence of populations as environments change and become more unpredictable[1,15]. Our results show that ostrich populations harbour individual variation in resilience to temperature change that may facilitate responses to shifting climates. However, this raises the question of why some individuals are more susceptible to temperature change than others? Given the fitness benefits of increased thermal tolerance, why has selection not eliminated variation within populations[57]? One possibility is that there are alternative strategies to cope with temperature change during reproduction. If thermal tolerance is costly, tolerant individuals that reproduce across a wide range of temperatures (generalists)

may have comparable fitness to individuals that only reproduce under specific thermal conditions (specialists), if they have lower reproductive success per breeding attempt[1,58–60]. We found no support for this idea, and if anything the opposite was true: Certain females appeared to specialize in reproducing at higher temperatures by increasing both the number and mass of eggs they laid, with no apparent reductions in egg mass at other times. It is possible that the ability of females to increase laying rates without compromising egg mass under extreme temperatures is facilitated by their unique life-history characteristics, including laying extremely small eggs relative to their body size. Whether certain life-history characteristics increase or decrease the vulnerability of species to climate change is unclear and clearly warrants further investigation.

Another possibility is that variation in thermal tolerance is maintained due to alternative breeding strategies. Ostriches have an extremely flexible breeding system, reproducing in both pairs and cooperative groups[37–39]. Cooperative breeding in birds has been shown to be a successful strategy for coping with high and fluctuating temperatures where breeding in pairs often fails[61–63]. In this study, it was necessary to restrict breeding opportunities to pairs to gain detailed measures of individual reproductive success. It is therefore possible that the sensitivity of individuals to temperature change may be alleviated by the buffering effects of sociality when opportunities to breed in groups arise[64–67].

This study shows thermal stress is an important factor that can limit reproductive success (see also Nord and Nilsson[68] and Walsh et al.[9]), even in species, such as the ostrich, that are well adapted to survive in extreme thermal environments. To explain the past and predict the future effects of climate change, it is crucial to quantify the effects of temperature on the fertility in species inhabiting different biogeographical zones and with different breeding biology. The extent to which the results of this study can be generalized remains to be established, given that little is known about temperature-dependent fertility in other tropical and sub-tropical species. However, the challenges faced by endotherms in arid, tropical and sub-tropical regions are clear and have already led to the collapse of entire bird communities[34]. A key feature of climate change highlighted by our results is that both hot and cold temperatures likely pose a challenge for species, providing an illustration of why temperature fluctuations, and not just temperature increases, are critical to study.

## Methods

**Study site and population**. The study site is situated at the Oudtshoorn Research Farm in the arid Klein Karoo of South Africa (GPS: 33°38′21.5″S, 22°15′17.4″E). The ostriches used in this study are derived from 139 founding individuals, consisting of individuals classified into one of two subpopulations with the popularized names South African Blacks (*S. camelus*) or Zimbabwean Blues (*S.c. australis*). From 1998 to 2018 the reproduction of captive breeding pairs ($n_{females} = 756$, $n_{males} = 701$) was monitored in 197 enclosures of ~0.25 ha of natural Karoo habitat[69]. A male and a female ostrich were assigned to each enclosure in May/June each year and kept together until the end of the breeding season in December/January. Male–female combinations were established to prevent inbreeding and where possible, generate new combinations each year. From 2008 to 2018 the fertility of males ($n = 22$) kept in solitary enclosures (20 m × 17 m) and trained to ejaculate into an artificial cloaca using a dummy female was monitored (method developed by Rybnik et al.[70]). Ostriches received a diet designed for breeding individuals (90–120 g protein, 7.5–10.5 MJ metabolizable energy, 26 g calcium and 6 g phosphorus per kg feed) and water ad libitum. Levels of dietary protein and energy were reduced across years to lower feed costs, which had negligible effects on fertility[71,72]. Maximum daily temperature records were obtained from a local weather station 600 m from the study site. Ethical clearance was obtained from the Western Cape Department of Agriculture (DECRA R12/48).

### Reproductive data

*Female gametic traits*. Pairs were checked twice a day and any eggs were collected and weighed using an electronic balance (Mercer). This gave us an estimate of the daily changes in quantity and mass of female gametes, that could be directly compared to daily temperatures. In two years the laying season was extended beyond February until April. All data from these months were removed to ensure data were consistent with other years. We also removed data from pairs where the male or female was replaced during the breeding season, which occurred sometimes when individuals were injured or died. Data on the rate of egg laying from these replacement pairs indicated that acclimation to enclosures and new partners takes ~45 days (Supplementary Fig. 5). Based on this information we removed data from the first 45 days from each season. Two-year-old females had substantially lower reproductive success than older breeders (Supplementary Fig. 6, see also Cloete et al.[69]) so these were removed from the data. Pairs that spent fewer than 200 days in their enclosure in a given year were removed so that data were consistent across pairs and years. Finally, pairs that laid fewer than 10 eggs per year were removed to avoid including incompatible pairings and individuals not in breeding condition, which reduced the total number of females in the analyses to 652.

*Male gametic traits*. For males, the ability to deliver high quantities of sperm of high quality is crucial for fertilization success[73–75]. We obtained natural ejaculates from solitary males kept in individual enclosures and estimated the number of sperm and sperm viability. Semen collections were performed three to five times a week and after periods of sexual rest the first three ejaculates collected were discarded. From the resulting set of ejaculates we kept data on the first ejaculate collected each day, typically obtained in the morning, from each individual. Sperm concentration was measured with a spectrophotometer in 20 μL semen diluted 1:400 (v/v) with a phosphate buffered saline solution containing 10% formalin. The number of sperm was estimated as the product of sperm concentration and ejaculate volume, which we estimated using an automatic pipette. Sperm viability was estimated by inspecting 500 sperm stained with nigrosin-eosin, and characterizing a sperm as viable if the morphology was normal (complete unit of tail, midpiece and slightly curved head)[76], the membrane was intact and eosin impermeable[77]. Only males from which we were able to obtain at least five ejaculates were included in the analysis to avoid including males not accustomed to the ejaculation collection process. Subsets of these data have previously been used to test effects of season, age and collectio[78–80].

*Hatching success and number of offspring*. Hatching success reflects the product of both male and female gametic traits as well as the quality of incubation. To control incubation effects, eggs were artificially incubated in an on-site hatchery until hatching. Eggs were stored (1–6 days) at 17 °C and 80–90% humidity with two daily rotations through a 180° angle until eggs were moved to incubators once a week. Eggs were incubated at 36.2 °C and 24% humidity with hourly rotation on their long axis through a 60° angle for the first 35 days and then switched to a hatcher set at 36 °C and 24% humidity for the remaining 7 days[81]. This dataset was subject to the same filtering procedure as the female egg traits.

### Statistical analyses

*Time lag effects of temperature on gametes*. The time period where different traits are influenced by fluctuations in ambient temperatures (i.e. the critical thermal window) is unknown. We therefore estimated the sensitivity of each trait to different sliding thermal windows preceding gamete production using general linear models (GLMs), where different thermal windows were entered as predictors of gametic traits at the population level. A window size of 3 days was chosen and one day steps were examined from 7 days before to 5 days after egg laying. We chose a window size of 3 days to capture immediate temperature fluctuations, while minimizing the effects of seasonal trends that occurred with larger windows. This also enabled us to avoid missing daily extreme events that occurred with smaller windows. Supplementary one-day and two-day window analyses supported this decision, as three sequential one-day windows (or two overlapping two-day windows) were particular important predictors of egg-laying (Supplementary Fig. 7). The thermal windows after egg laying served as controls, as we did not expect any predictive power apart from the autocorrelation in temperature. In each window, the average daily maximum temperature (AVG-$T_{MAX}$) was modelled as a quadratic effect. To identify the critical thermal window, we compared the models using Akaike information criterion (AIC) or QAIC (Quasi-AIC) to account for the overdispersion common to logistic regressions. The maximum egg-laying rate is one egg every 2 days. We therefore modelled the probability of laying as the number of 2-day intervals with (eggs/2 days) and without eggs using a Binomial error distribution, which was necessary to correctly model the variance in successes (our response ranged from 0 to 1 whereas eggs per day ranged from 0 to 0.5). Model comparison with QAIC showed that the critical thermal window was 2–4 days before egg-laying (Supplementary Fig. 1). Interestingly 2 days is also the time it takes for eggs to travel down the oviduct[82,83]. Egg mass was modelled using a Gaussian error distribution and the ranking of AIC was very sensitive to small model adjustments and extreme temperatures, reflecting the generally low effect of temperature on this trait (Fig. 1) and see the section "Discussion"). Visual inspection revealed a consistent trend of increasing egg mass at extreme high temperatures but not at intermediate to high temperatures (Fig. 1). To reduce the influence of these extreme data points, without removing the entire trend of what may be a true biological signal we removed the 0.5% hottest and the 0.5% coldest records in this particular analysis. Several thermal windows prior to egg-laying appeared to predict egg mass equally well, but we proceeded with 0–2 days before egg laying as

the critical window for this trait due to its proximity to day of laying (Supplementary Fig. 1). For both hatching success (Binomial error distribution: number hatched vs. number not hatched) and the number of offspring (Binomial error distribution: 2-day intervals with chicks vs. 2 days without chicks, chicks/2 days) we used 0–4 days before egg laying as the critical thermal window as this included all days used as predictors for egg mass and egg laying rate. In birds, spermatogenesis is believed to range from 11 to 15 days[84], and we therefore tested thermal windows from 15 days before to 5 days after ejaculation. The critical thermal window for the number of sperm (Poisson distribution) was 2–4 days before ejaculation, and while sperm viability (Binomial error distribution: number alive vs. number dead) was also influenced by temperature during this time, the window 4–6 days before ejaculation was a better predictor (Supplementary Fig. 2). However, as results did not differ between the analyses of sperm viability detailed below (random regression and character-state models) when using 2–4 vs. 4–6 days, we used 2–4 days for consistency across traits. The critical thermal windows estimated for sperm and egg traits are specific to this study. If other species are studied it will be important to estimate these parameters using similar critical thermal window analyses from time series datasets.

*Thermal stress index.* For each trait we modelled the response to increases and decreases in temperature by creating cold and heat thermal stress indexes. This was done by first estimating the temperature at which trait values were maximized (thermal optimum), and secondly by calculating decreases ($T_{cold\ stress}$) and increases ($T_{heat\ stress}$) away from this optimum. Using GLMs we modelled the change in number of sperm and eggs produced as a response to AVG-$T_{MAX}$ (linear and quadratic terms) of the critical thermal window, and extracted the parametric vertex as the thermal optimum (rounded to closest degree Celsius). For egg laying the optimum was estimated as AVG-$T_{MAX}$ 20 °C (Supplementary Fig. 3), which also reflects the centre of the TNZ of the emu[40] (unknown for the ostrich). For the number of sperm ejaculated the optimal temperature was estimated to be 26 °C (Supplementary Fig. 4). As a result, $T_{heat\ stress}$ for females was from 20 to 45 °C and for males it was from 26 to 45 °C. $T_{cold\ stress}$ was from 20 to 10 °C for females and from 26 to 10 °C for males. The observed difference in thermal optima between sexes is intriguing, but this dataset was not designed to robustly test for sex differences: the fitness of males and females are intertwined in the pairs and we have no direct data on how solitary male sperm performance influenced female fitness. To make the intercept of the statistical models represent the most benign temperature we subtracted the minimum stress value resulting in 0 being the no minimum (no stress) of the thermal stress index. The variance of slopes (see below) depends on the scale of the environmental parameter and we therefore standardized this by dividing by the maximum of the range resulting in 1 being the maximum deviation from 0.

*Modelling resilience to temperature change using random regression models.* We constructed random regression models in R v.3.6.0[85] using the Bayesian framework implemented in the R-package MCMCglmm v.2.29[86]. For both residual and random terms we used the weakly informative inverse-Gamma distribution (scale = 0.001, shape = 0.001, i.e. $V = n$, $nu = (n−1 + 0.002$ with $n$ being the dimension of the matrix) as priors. For female gametic traits, models were run for 10,000,000 iterations of which the initial 100,000 were discarded and only every 10,000th iteration was used for estimating posterior probabilities. For male gametic traits, models were run for 3,000,000 iterations, of which the initial 30,000 were discarded and only every 3000th iteration was used for estimating posterior probabilities. The number of iterations was based on inspection of autocorrelation among posterior samples in preliminary runs. Convergence of the estimates was checked by running the model three times and inspecting the overlap of estimates in trace plots and the level of autocorrelation among posterior samples. Posterior mode and 95% credible intervals are reported for random effects, correlations and repeatability measures. Models included the fixed effects of thermal stress (ranging from 0 to 1) and stress type (cold or heat). The interaction between thermal stress and stress type was modelled with a common intercept for cold stress and heat stress, as the construction of the thermal stress index dictated that these intercepts are identical.

For the three traits modelled with Binomial error distributions (egg laying, hatching success and number of offspring) data were grouped into four hot and three cold thermal stress classes, each representing the number of observations with success (e.g. 2-day intervals with egg) and the number of observations with failure (e.g. 2-day intervals without egg). For female gametic traits we included the additional fixed effects of female subpopulation (South African Blacks: 476 females, Zimbabwean Blues: 68 females or crosses: 108 females) and its interaction with the thermal stress and stress type, as well as female age and the subpopulation of the pair male. Results were highly consistent across subpopulations and we therefore report fixed effect estimates from the most numerous subpopulation (South African Blacks) for brevity. Population-specific estimates are available in the results tables provided in the supplementary information. The mass of eggs decreased with the number of days since the previous egg (Supplementary Fig. 8). This was accounted for by including days since previous egg (linear and quadratic terms, log-transformed) as a fixed effect in the egg mass model. Several sperm-characteristics may peak at an intermediate age[78], and therefore linear and quadratics effects of age were included as fixed effects in models. We accounted for environmental effects that differed across years, such as diet, by including year as a

random effect. For egg-laying rates, egg mass, hatching success and offspring number, enclosure was also added as a random effect, since they were used repeatedly across years and varied in size and vegetation cover. The males used for sperm collection were kept in the same enclosures across years and therefore we did not have enclosure as random effect in analyses of sperm traits (not possible to separate individual from enclosure effects). The enclosures where males were kept for sperm collection are, however, extremely similar making it unlikely that this was a significant source of error variance.

*Quantifying individual variation in resilience to temperature change.* In all models the thermal stress index and type of stress (cold versus heat) was allowed to interact with ostrich ID (id) to model the individual variance (id). This was modelled as $3 \times 3$ unstructured variance–covariance matrix composed of the intercept ($id_{int}$), slope during cold stress ($id_{sl-cold}$) and slope during heat stress ($id_{sl-heat}$). Individual repeatability ($R$) of trait values at the optimum temperature ($T_{stress} = 0$, 20 °C for females and 26 °C for males) was then estimated as the proportion of intercept variance that is explained by the individual variance in intercepts:

$$R_{int} = \frac{\sigma^2_{id_{int}}}{\sigma^2_{id_{int}} + \sigma^2_{year} + \sigma^2_{enclosure} + \sigma^2_{res}}. \quad (1)$$

Individual variation in the cold and heat stress slopes was used as an estimate of variation in resilience to increasing and decreasing temperatures, i.e. phenotypic plasticity. However, to estimate the repeatability of slopes for individuals (consistency of individual by environment interaction; I × E), we constructed a second set of models. In these models a second $3 \times 3$ unstructured variance–covariance matrix of individual by year (id-yr) combinations was added, allowing the repeatability of thermal plasticity within individuals across different breeding years to be calculated. Variance in individual slopes is on a different scale to that of intercepts, and also dependent on the scaling of the temperature index. For these reasons we followed a recently introduced practice[87,88] and estimated the repeatability of thermal slopes as the proportion of slope variance attributable to between individual variance:

$$R_{sl} = \frac{\sigma^2_{id_{sl}}}{\sigma^2_{id_{sl}} + \sigma^2_{id-yr_{sl}}}. \quad (2)$$

To quantify how much variation in each trait was explained by responses to temperature we transformed the between individual and within individual slope variance to the same scale as the intercept variances using $\sigma^2_E = \sigma^2_{sl} * var(x)$, where $var(x)$ is the variance of the environmental variable, the temperature index[89]. We then expressed this variation as a ratio of the total variance, including between individual and within individual intercept variance as well as year, enclosure and residual variance:

$$PSlopeVar = \frac{\sigma^2_{id_{Ehot}} + \sigma^2_{id-yr_{Ehot}} + \sigma^2_{id_{Ecold}} + \sigma^2_{id-yr_{Ecold}}}{\sigma^2_{id_{Ehot}} + \sigma^2_{id-yr_{Ehot}} + \sigma^2_{id_{Ecold}} + \sigma^2_{id-yr_{Ecold}} + \sigma^2_{id_{int}} + \sigma^2_{id-yr_{int}} + \sigma^2_{year} + \sigma^2_{enclosure} + \sigma^2_{residual}}. \quad (3)$$

It has recently been debated if the fixed effect variance ($\sigma^2_f$) should be included in the denominator when estimating $R$[90]. There are arguments for including fixed effect variance if it captures natural variation and excluding it if it represents experimental variance[91]. For full transparency we chose to report estimates of $\sigma^2_f$ excluding variance from the thermal index ($\sigma^2_{f-thermal\ stress}$) as this parameter has already been accounted for by the interaction with the random terms. We estimated fixed effect variance of all terms ($\sigma^2_{f_{all}}$) and of thermal stress separately ($\sigma^2_{f_{thermal\ stress}}$) following de Villemereuil et al. [91], such that $\sigma^2_{f-thermal\ stress} = \sigma^2_{f_{all}} - \sigma^2_{f_{thermal\ stress}}$.

As egg laying, hatching success and number of offspring are modelled via a logit link function, estimates of $R$ are calculated on the latent scale. While this scale has the benefit of fulfilling the typical assumptions of parametric analyses, it may not reflect the scale at which selection is working. Methods have therefore been developed to make inferences on the observed scale[92]. There are currently no methods to perform this transformation for a model using a logit-link function and where the number of trials varies between data points. Instead it is possible to calculate estimates of repeatability on the expected scale (corresponding to the liability scale in a threshold model) according to equations in de Villemereuil et al. [92] using the R-package QGglmm[92]. Similar methods are not available for the slope variance parameters presented below, and all estimates presented in the main document are therefore on the latent scale for consistency. Where possible, we also provide estimates on the expected scale in the supplementary material (Supplementary Tables 2–10).

*Modelling resilience to temperature change using character-state models.* As an alternative modelling approach to random regression, we modelled changes in each trait across three temperature categories (cold, benign and hot), using character-state models. For egg-related traits the ranges for these categories were limited by the lower number of cold days compared to hot days, according to the thermal optimum cut-off used in the random regression analysis (20 °C). To avoid low replication in the cold category relative to hot days we assigned the lowest 50% of days classified as $T_{cold\ stress}$ as cold (<17.7 °C, $n_{eggs} = 10,483$) and the highest 30% of days classified as $T_{heat\ stress}$ as hot (>28.7 °C, $n_{eggs} = 14,759$), with the remainder being classified as benign ($n_{eggs} = 56,297$). Data on sperm traits had higher temperature

values. We therefore increased the temperature cut-offs (cold: <18.7 °C, $n_{ejaculations}$ = 319; hot: >29.7 °C, $n_{ejaculations}$ = 392 and benign $n_{ejaculations}$ = 1174). The models were constructed in MCMCglmm v.2.29[86] and followed the same general approach as the random regression models described above. The major difference was that temperature category was included as a fixed factor and the interaction between the random effect ostrich ID and temperature category was modelled as a $3 \times 3$ unstructured variance–covariance matrix composed of the cold, benign and hot temperature categories. We also estimated the residual variance separately for each temperature category (see Supplementary Tables 12–17 for further details on the model components).

*Modelling trade-offs between traits.* To quantify correlations between female gametic traits (egg mass vs. number of eggs with 0–4 days before egg laying as the critical thermal window) and between male gametic traits (sperm viability vs. number of sperm with 2–4 days before ejaculation as the critical thermal window) two-trait models were used. These were setup using MCMCglmm v.2.29[86] with the same error distributions as the single-trait models. For female gametic traits, models were run for 5,000,000 iterations of which the initial 100,000 were discarded and only every 2000th iteration was used for estimating posterior probabilities. For male gametic traits, models were run for 3,000,000 iterations, of which the initial 30,000 were discarded and only every 3000th iteration was used for estimating posterior probabilities. Each trait comparison was analysed with both random regression models and with character-state models, containing the same fixed effects as the single-trait models, but with the reserved term "trait" interacted with all fixed effect components. Models also contained the same basic random effects as the single trait models, but with the random effects and residuals estimated separately for each trait. In the random regression models the interaction between ostrich ID and thermal stress was modelled by constructing two $4 \times 4$ unstructured variance–covariance matrices, one for $T_{heat\ stress}$ and one for $T_{cold\ stress}$, composed of the intercept and slope for both traits. Two similar matrices were constructed for the interaction between individual by year (id-yr) records and thermal stress. Using these matrices, we extracted covariance between traits in the response to heat or cold stress among and within individuals, which was then used to estimate correlations (correlation = covariance$_{trait1,trait2}$/sqrt(var$_{trait1}$*var$_{trait2}$)). In the character-state models the interaction between ostrich ID and temperature category was modelled by constructing three $2 \times 2$ unstructured variance–covariance matrices composed of either the cold, benign or hot thermal category for both traits. These matrices were used to extract covariance components between traits among individuals for a given thermal category, and use these to estimate correlations. Similar matrices were also used to model the residual variance (within individuals) in the character-state models.

**Reporting summary**. Further information on research design is available in the Nature Research Reporting Summary linked to this article.

## Data availability
The data that support the findings of this study are available from the Western Cape Department of Agriculture in South Africa (WCDA). Restrictions apply to the use of these data, and so are not publicly available. Data are however available from the WCDA upon request. Source data are provided with this paper.

## Code availability
The code used in this study is available as an R-file (Supplementary Code 1).

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

## Acknowledgements

We are thankful to all the staff and workers at Oudtshoorn Research Farm for assistance with data collection and maintenance of the birds and to the Western Cape Government

for use of their resources. Special thanks to Naomi Oosthuizen for assistance with semen collections. The computations were performed on resources provided by SNIC through Uppsala Multidisciplinary Center for Advanced Computational Science (UPPMAX) under Project SNIC 2018/8-359. This research was funded by a stipend from the Carlsberg Foundation to M.F.S. and by the Swedish Research Council (grant number 2017-03880), the Knut and Alice Wallenberg Foundation (Wallenberg Academy fellowship number 2018.0138), Carl Tryggers (grant numbers 12: 92 and 19: 71) and an Oppenheimer travel exchange fund to C.K.C. E.I.S. was financially supported by a grant from the Swedish Research Council (grant number 2016-03356). The maintenance and development of the ostrich populations used in this study was funded by the Western Cape Department of Agriculture and supported by grants from the Western Cape Agricultural Research Trust (Grant number 0070/000VOLSTRUISE: (Cloete)) as well as the Technology and Human Resources for Industry programme (THRIP—Grant number TP14081390585) of the South African National Research Foundation to S.W.P.C.

## Author contributions

Conceptualization: M.F.S., C.K.C.; Data curation: M.F.S., C.K.C., M.B., A.E., Z.B., P.T.M., S.W.P.C.; Formal analysis: M.F.S.; Funding acquisition: M.F.S., S.W.P.C., C.K.C.; Investigation: M.F.S., M.B., A.E., Z.B., J.M., P.T.M., S.W.P.C., C.K.C.; Methodology: M.F.S., E.I.S., C.K.C.; Project administration: M.F.S., M.B., C.K.C.; Writing—original draft: M.F.S., C.K.C.; Writing—reviewing and editing: M.F.S., M.B., A.E., Z.B., E.I.S., J.M., P.T.M., S.W.P.C., C.K.C.

## Funding

## Competing interests

The authors declare no competing interests.
