## [Peer Review File · Nature Communications]

Reviewers' Comments:

Reviewer #1:

Remarks to the Author:

A recent emphasis on the potential of climate change to negatively impact reproductive processes, and not only survival, motivates this study. This paper examines the role of natural fluctuating temperature (stress) in a captured population of ostrich on reproductive success, measuring the consequences of both hot and cold stress among and within males and females. Ejaculate traits of males (sperm number and viability) and egg traits of females (egg number and mass) were measured and then tested for the effect of temperature fluctuations at the site. Data represent traits collected over XX years. The results show that gamete number (sperm and egg number), but not gamete quality (sperm viability and egg mass) decline under both cold and hot stress. There is individual variation in this response, suggesting there is capacity for the population to respond to future thermal variation. No tradeoffs were found between reproductive traits and the authors suggest that population level variation in the consequences of thermal stress on reproduction is maintained by individuals specializing in responses to hot temperatures.

I found the ms. exceptionally well-written, easy to understand and mostly well-justified. The work is novel and will be of interest to others in the community and wider field as this represents a substantial dataset, collected from a semi-natural population, addressing a question that is of increasing importance in the wider field of climate change biology. I found the statistical techniques used throughout the ms. to be mostly easy to follow (although I am unfamiliar with several of the approaches) and will serve as a good model for other researchers in using the data available to construct for example thermal windows, sensitivity to temperature, etc. Overall, a clever approach with the – by default – restricted types of data that can be collected from wild populations. The authors did exclude some data but these exclusions were based on the biology of the birds and justified statistically.

I have several queries for the authors that may improve the paper from both the perspective of clarity but also strength of the results.

1) Much of the results and subsequent impact of the paper are shown in Fig. 2 and, in particular, panel F. I have several issues regarding this figure.

a. The legend provides N, eggs = 81297 and Nobs sperm = 1785. However, these are misleading because this value should represent the number of individuals measured. The finite number of individuals in the population means these values are repeated measures for females and females. I trust that all the R models incorporated individual as a random effect to control for this? Assuming that to be the case, the N values in the panel are overinflated and should be changed to reflect the number of individuals.

b. Panel F shows one of the most important results but I found it difficult to find in the methods the justification for the Y axis variable, "chicks/2 days". How sensitive is this main result to variation in the Y axis variable? I would encourage the authors to be more clear in the justification for this in the main text and to refer in the main text to the area in the methods where this justification is statistically confirmed.

c. Related to panel F, I think it is very important for the authors to emphasis in the main text that incubation quality cannot explain this effect, since as it states only in the methods currently, that incubation is done artificially and therefore incubation quality is controlled for.

2) While I appreciate that the goal of the work was to assess individual reproductive success and its sensitivity to natural thermal fluctuations, the idea that cooperative breeding may offset individual consequences is interesting. While the authors state on line 253-254 the study was restricted to breeding pairs, its unclear whether cooperative breeding is allowed on the site. If so, then there are several ways the authors can test the hypothesis that sociality buffers the population from individual reproductive consequences of thermal variation. Perhaps one way to do this would be to compare the

outcome of episodes of cooperative breeding with paired breeding using the character state modelling approach (e.g., cold, benign, hot). Obviously because the authors understand the structure of their data better than me, I'm sure the authors can be clever in testing this hypothesis. While the work stands by itself without it, the impact of the work may be improved by formally testing whether flexible breeding systems could buffer climate change using data collected simultaneously on the reproductive outcome of paired and cooperative breeding attempts in a semi-wild population undergoing natural thermal variation. How cool! (or hot...)

3) I found it very interesting that there appears to be a mismatch between egg laying peak and thermal optimum for sperm number. The authors do not discuss the potential consequences of this. I find this surprising since the authors also state that male reproductive success in ostriches is contingent on high quantity and quality of ejaculates. Thus female reproductive optima is at a cooler temperature than male reproductive optima. Yet there are also females that specialize in reproduction at hotter temperatures. Do the authors have anything to say on this potential sexual conflict and female variation?

4) Minor comments:

a. Line 243 has a referencing issue. Gilchrist...

b. Line 303 requires a reference to justify the statement that male reproductive success in ostriches to deliver high quantity and quality ejaculates.

c. Line 347 appears to have a typo: with without

Reviewer #2:

Remarks to the Author:

This is a really interesting study, with a great dataset, investigating important questions about how reproductive traits are impacted by changes in temperature and how much this varies across individuals. I think it will be of broad interest to those interested in both evolutionary and climate change biology. The methods appear to be thorough and robust, and the conclusions drawn are mostly fair and well considered. The paper is nicely written and easy to follow, and does a good job of putting these results into the wider context of how fluctuating temperatures may impact bird reproductive success as the global climate continues to change. I do have a few questions for the authors and some suggestions for improvement, but I think these can mostly be addressed by adding some more information/explanation to the text and making some relatively minor amendments. I'll first mention two broad points and then go through more specific comments on a line-by-line basis.

Broad comments

My first broad comment is that I am not convinced of the premise, laid out in the introduction, that the effects of temperature change on reproductive success is little studied in birds – in fact, there are a number of studies that have addressed this question in different ways (e.g. Järvinen 1994 *Ecography*; Kitaysky & Golubova 2001 *J. Anim Ecol*; Julliard et al. 2004 *PRSB*; Weatherhead 2004 *Global Change Biology*; Auer & Martin 2012 *Global Change Biology* – list is not exhaustive), as well as a wealth of poultry science studies showing such effects experimentally. This doesn't detract from the current study, but rather places it within a growing and important topic that is of broad interest.

My second broad comment is that I think it should be recognised, in the discussion and perhaps in the abstract/intro too, that ostriches are a special type of bird, and several ecological and physiological factors important for them may not be applicable across birds more broadly. For example, ostriches are already well adapted for breeding in hot arid conditions (they could be considered already relatively 'heat-stress adapted'), and indeed several characteristics of their eggs help them cope with such conditions. Ostriches also have relatively small eggs for their body size and therefore relatively low egg investment, which may mean they are less likely to be subject to intense trade-offs between egg size and number, even under stress. The paper attempts to make some broad conclusions about

thermal stress representing a "considerable and underappreciated factor limiting avian reproductive success", but I would like to see these conclusions tempered slightly to acknowledge that these issues could (and may be likely to) be quite species-specific. Again, rather than detracting from the study, this point actually highlights the importance of species-specific studies like this.

Specific comments

line 19: "(-5°C - 45°C)" this should perhaps be changed to something like "below X°C and about Y°C" since you are referring to extremes but this range includes 'normal' temps

line 20: I don't see that there's any immediate reason to expect an effect of elevated ambient temperature on gamete quality, since all bird sperm are produced inside the body at high temps

line 34: it is important to distinguish here between fertility specifically (i.e. effects on a focal individual's ability to produce functional gametes – which is what you appear to be testing) and prenatal death of a zygote/offspring. Some of the papers referred here don't actually differentiate the effects of temperature on traits that affect fertilisation success/failure from those that affect surviving offspring produced ("fertility" can be used as a catch-all term). Failure to distinguish these two things will mask the mechanistic basis of any temperature-mediated impacts on reproductive success. This distinction will help to inform several of your other hypotheses too.

line 55: a recent study by Hällfors et al (2020, PNAS) showed that breeding periods are often not just advancing but also contracting – this may have important implications for reproductive outputs, especially for species that produce >1 clutch or frequently fail on first attempt and re-lay

line 69: this is a really great dataset

line 74: reproduce repeatedly across the year in response to what? is onset of breeding directly/indirectly driven by temperature?

line 76: are all the birds in this study captive? I.e. all 1299 birds studied (line 71). It would be useful to clarify if this is a wild, semi-wild/managed (if so, how?), or captive population in the main text, to help the reader fully understand the system set up

line 77: are the solitary males also in enclosures? Are they kept alone? Being alone and isolated from other individuals (both male for competition and females for sexual arousal) may influence sperm production.

line 95: sperm number is likely to be influenced by how frequently males have recently ejaculated – presumably this will differ between paired and unpaired males (and perhaps between different paired males too, depending on how consistent copulation rates are across pairs)? Lack of ejaculation could also influence sperm viability and this could be a particular problem for solitary males – fewer ejaculates = older deteriorating sperm? Was sperm collected from solitary males regularly to account for this? This is not clear from the methods section either.

line 109: although egg mass was unaffected, other egg traits e.g. egg composition may have varied, without necessarily affecting overall mass/size. This can also be important for hatching success. A brief point on this would be useful in the discussion.

line 111: I think this is no surprise really, so perhaps not "remarkable" – as mentioned earlier, in birds the testes are inside the body, so sperm are made in very warm conditions! I don't think there is much argument here for sperm quality being directly impacted by ambient temperature. Perhaps for ambient temperature affecting other aspects of the male's physiology, with knock-on effects for sperm production (but in that case, we might expect it to be numbers rather than sperm morphology). I'd

like to see this point clarified in the discussion. Linked to this: would we necessarily expect the impact of higher vs lower temps on gamete production to be the same? Mechanistically, surely not.

Fig 2: the deviation range for decreasing temps seems much narrower than the range for increasing temps. Why is the shift from the optimum not consistent across plots (are different data used for each?). Looking at your methods, it appears that this is probably explained by sperm being sampled over a more limited (but still impressive!) period of time - if this is correct, please spell out the sample sizes for the different analyses more clearly.

Fig 2: Is (F) needed? If hatching success is not strongly influenced by temp, then doesn't the decline in offspring (F) simply reflect the decline in egg numbers (A)?

line 145: I'm not sure it is valid to conclude "cumulative detrimental effects on eggs and sperm" - yes for eggs, but why sperm? Egg numbers decline and this drives the decline in offspring number. It seems to me that the effect on hatching success is limited and since developing embryos are highly sensitive to temperature and associated humidity changes, any impact on hatching success could be direct effect on embryo survival (i.e. nothing to do with fertility or sperm). Developing eggs are particularly sensitive to humidity fluctuations (even during artificial incubation, since many artificial incubators do not stand up well to extreme ambient conditions when auto-regulating temp/humidity).

line 149: these are such interesting results – it's as though there may be 'subpopulations' of individuals with slightly different trait profiles. Would be fascinating to know what the mechanistic drivers are (I hope to see this in a future study!)

line 162: I wonder if this is because egg size is relatively small in ostriches (for their body size), so there is not as much constraint and it is relatively easy to maintain egg size even under stress? Maybe you would see more dramatic effects in a bird with large relative egg size, as this egg requires relatively more investment by the female?

line 168: I think egg size (and shape) is probably mostly determined (and constrained) by the size/morphology of the female reproductive tract, so this makes good sense to me

line 174: are these fertility related traits always depressed at the individual level when temperature deviates from the optimum, or do some individuals specialise to the extent that they 'improve' with slight increases/decreases in temp? This is what your phrase "some females appeared to specialise at reproducing at high temperatures" (in abstract, line 21) implies, and if so it's very interesting indeed and I would like to see the point be made more clearly.

line 192: this is another example where what we see in ostriches may not be so easily applied to other birds. Temperature stress may typically play a very important role in inducing trade-offs between egg numbers and egg mass in a species where producing large eggs is relatively hard work/high investment. As mentioned above, ostriches have relatively small eggs for their body size, so relatively low investment, and this may mean it's just not such a big deal for them.

line 205: was it six adult individuals that died from heat stress? What about embryos/chicks/juveniles (can you tell)?

line 208: cooler temps are likely to be more of a problem for species already adapted to hot environments (like ostriches). I also (again) wonder about what triggers the onset of reproduction in this species, given that they breed opportunistically throughout the year. Is it known why they choose certain times to breed? Temp may be an important mechanistic trigger (proximate cause).

line 221: I think you should clarify that there is no sign of an upper thermal limit *within the natural range of temperatures that the birds encountered during the study* (there will, of course, be a limit)

line 222: how was egg number was adjusted? Did females lay smaller clutches, or fewer clutches, or stop laying sooner, or widen gaps between eggs... again, wondering about the mechanisms that underpin these effects

line 228: again, it's also worth noting that these gamete production processes already take place at much higher temperatures in birds than most other animals, because body temp is already high (~40°C in passerines – I don't know if that is true for ostriches!) and testes internal in birds. Sperm production in birds *must* already be adapted to high temps relative to most other species. All the references listed here are mammalian studies, where testes are relatively external and therefore cooler.

line 245-246: this answers my earlier question about whether some females actually did better in elevated temps: it's such an interesting finding, but I didn't feel it was entirely clear from your results (to me at least). Can you make this clearer (is it possible to visualise in some way)?

line 261: it is crucial to quantify the effects of temp on fertility of species in different biogeographical zones, but I would also argue that we must consider how this may differ across different reproductive strategies e.g. it might not be too bad if you are an opportunistic breeder and can wait for conditions to be 'right' but if you are a seasonal breeder, suboptimal temperatures during the breeding season could be extra costly. In addition, as said before, in species where there may already be a stronger trade-off between reproductive traits e.g. where egg size is relatively large and therefore a bigger investment, the impacts of temp might extend to gamete quality as well as quality.

line 269: am I inferring correctly from this that the ostriches are research/farmed animals, not wild? It is important to make this clear both here and at the relevant point in the main text. The fact they have ad lib food, water, etc etc as a result of being captive has the potential to make a big difference and is something that should be addressed in the discussion (the impacts of temperature on reproductive success could be far more dramatic in wild systems)

line 276: please provide sample size of males from which sperm samples were analysed here, and above please provide sample size of breeding pairs monitored (which presumably = number of females analysed for egg production?)

line 300: please provide final sample size for females analysed after all this data removal

line 305: were the males routinely ejaculated at regular intervals on the run up to collecting the experimental samples? Wondering about how 'fresh' the sample was (males can store old sperm for some time if they don't have the opportunity to ejaculate regularly)

line 311: I'm not sure this is convincing with respect to overall 'viability' – sperm that appear morphologically normal are often non-motile, sperm with non-intact membranes are sometimes motile; these assays are not a perfect measure of sperm viability and fertilising ability. I would prefer for the specific sperm traits to be referred to both here and in the main text, to avoid misleading the reader i.e. temperature didn't affect sperm morphological normality or sperm membrane integrity. Until I got to the methods, I was assuming some sperm motility analysis or similar had been performed.

line 315: analysing failed eggs for fertilisation success would have been a better measure of sperm function/success, because it is the only way of demonstrating that the sperm actually made it to the egg. This is why when you are talking about hatching failure in the main text you cannot conflate it with "infertility" – it could be embryo mortality

line 333: I agree that thermal windows after egg laying should not affect sperm or egg production, but

could they not influence hatching success e.g. if elevated (or reduced) temperatures are more likely to result in early embryo death (which lots of poultry science literature suggests they are)? See Christensen 2001 World's Poultry Science Journal for a review.

lines 326-355: is anything known about what is happening physiologically (i.e. inside the bird) with respect to gamete production during these critical periods that have been identified using this method? It strikes me that it might be important to consider what the critical phases in gamete production might be, and that this should be fed into this somehow.

Title: I would prefer "fertility" to be replaced with "reproductive traits" since this is more accurate (the study doesn't explicitly measure fertility in terms of the ability of males/females to produce fertilised eggs).

Nicola Hemmings

Reviewer #3:

Remarks to the Author:

Using two decades of data, the authors investigate whether extreme temperatures have damaging effects on different fertility traits, explore individual variation in resilience of fertility to changing temperatures, and test whether extreme temperatures cause trade-offs in investment across traits. The novelty of this study comes from the fact that the authors are using a powerful dataset to test how temperature fluctuations affect fertility traits, rather than survival, and working in a tropical/sub-tropical system, rather than a temperate system. Interestingly, the authors find that only some fertility traits are sensitive to temperature extremes, and that some females appear to specialise in reproducing at higher temperatures, which could facilitate responses to climate change - a finding that will certainly influence thinking in the field. Overall, the manuscript is very well written, the statistical analyses appear to be robust, and the topic will be of broad interest to readers.

I only have a few minor comments:

1. How were the birds paired each year? Was it random? How often were the same individuals paired up over years? Or was this avoided?
2. What was the degree of relatedness? Were offspring from some years put into the breeding programme? If so, is it possible to look at whether offspring of resilient females are also resilient?
2. I could not really follow the sample sizes. There were 1299 individually marked birds across the 20 years, and data was collected from 650+22 (Figure 1B) of them? Further, I do not completely follow the n individuals by year = 80 in Fig 1 B, but then n = 18, 49 individuals by year combinations in the figure legend. I'd like to see a better breakdown of the sample sizes within and between years. This would allow to reader to understand how long individuals were followed and what the age structure was.

Reviewer #4:

Remarks to the Author:

This MS uses a very impressive dataset to evaluate the impact of temperature on fertility in ostriches, under semi-natural conditions. It finds that both male and female fertility are substantially impaired under naturally occurring temperature variation, by both cold and hot temperatures. I am unaware of any similar studies of this magnitude, and see this MS as a major advance in the field.

Overall, I am strongly in favour of publishing this MS. I think the dataset is compelling, and the conclusions reasonable and interesting. I have a two issues that I think need to be resolved, plus

several other points that are more optional but that I think will improve the MS.

Major issues (although I think they should be reasonably easy to resolve):

L96-98 and methods: there is no justification given for why a 3 day window is appropriate. Is there literature suggesting this is a good idea? Did you try other window sizes? I think you need to either show this doesn't matter, or provide evidence/literature that bird fertility is affected by windows like these. Quail spermatogenesis takes 13 days according to Lin & Jones (1992 doi:10.1007/BF00319382), and I can't see any reason to assume mature or nearly mature sperm should be particularly vulnerable to thermal stress, so limiting your range of potential windows to 7 days before mating also seems odd. It is entirely possible that mature sperm could be robust and early stage sperm are especially vulnerable. As far as I know, your work is the first to establish a time point for when thermal stress impacts on bird sperm- so in my view it is important to do it right. If you get this wrong, your paper will be used to justify a 3 day window by any other bird people interested in similar questions, which will cause problems for many other researchers.

I think your argument for thermal specialisation in females is fairly weakly supported by your data. I think you should tone down this argument (but still mention it- it is a very interesting idea). I think your MS is worth publishing even without this idea, so I would include it as an interesting possibility consistent with the data rather than the stronger version you give. Alternatively, convince me (and the other reviewer and editor) that you are right.

Minor issues:

L54 You are overstating this- there is a long literature examining the direct effect of temperature on sperm in mice, flies and livestock, for example. In general this paragraph is poor compared to most of your writing. It mashes ecological/phenology studies on birds (largely about food supply as you mention later) with temperate vs tropical. I suggest you rewrite this. 1) make a clear "most studies are on temperate species (or even specify temperate birds), so we need more information about tropical species potentially adapted to generally high temperatures". 2) A lot of work on birds has focused on adjustment of breeding time, which may be mediated by food availability, but there is also a need for more direct studies on sperm and egg traits which can also be directly affected by temperature at ecologically relevant temperatures (Hurley et al 2018). This is what you actually do, and it is worth contrasting how different this is to phenology.

L70 "in a tropical species, the world's largest bird, the ostrich (*Struthio camelus*)." makes it clearer you don't mean "the world's largest bird that lives in tropical countries" ie there are bigger birds in temperate zones.

L90 probably worth changing this to "Geographic range (green) of the" to make it as easy as possible for readers.

L91 "mean monthly range" could mean a wide variety of things (mean of the range of temperatures experienced in a month/mean of the max temperatures experienced in a month/mean difference between mean monthly temperatures in a year)- better to be really specific about what you mean.

Fig 2 I don't understand why you have ordered your graphs (A-F) completely differently to how you explain them in the figure legend (A, B, D, E, C, F).

L154-155 Wow!

Fig 3 Also wow!

L243 omit George W

Discussion- I think you need to briefly discuss whether ostriches are a good model for other organisms, or are weird. Do you expect similar effects in non-ratite birds? How about mammals as they are also endotherms?

L300 this would be stronger if you gave the mean number of eggs, or the number of pairs excluded vs included, to show this was a reasonable thing to do.

L340 it would be interesting to know if you think the difference in thermal windows in egg viability/size and male sperm number/viability indicated different biological processes and genuinely different vulnerable points, or if you think the viability and hatching data are unreliable because the effects are so much weaker

Tom Price

REVIEWER COMMENTS

Reviewer #1 (Remarks to the Author):

A recent emphasis on the potential of climate change to negatively impact reproductive processes, and not only survival, motivates this study. This paper examines the role of natural fluctuating temperature (stress) in a captured population of ostrich on reproductive success, measuring the consequences of both hot and cold stress among and within males and females. Ejaculate traits of males (sperm number and viability) and egg traits of females (egg number and mass) were measured and then tested for the effect of temperature fluctuations at the site. Data represent traits collected over XX years. The results show that gamete number (sperm and egg number), but not gamete quality (sperm viability and egg mass) decline under both cold and hot stress. There is individual variation in this response, suggesting there is capacity for the population to respond to future thermal variation. No tradeoffs were found between reproductive traits and the authors suggest that population level variation in the consequences of thermal stress on reproduction is maintained by individuals specializing in responses to hot temperatures.

I found the ms. exceptionally well-written, easy to understand and mostly well-justified. The work is novel and will be of interest to others in the community and wider field as this represents a substantial dataset, collected from a semi-natural population, addressing a question that is of increasing importance in the wider field of climate change biology. I found the statistical techniques used throughout the ms. to be mostly easy to follow (although I am unfamiliar with several of the approaches) and will serve as a good model for other researchers in using the data available to construct for example thermal windows, sensitivity to temperature, etc. Overall, a clever approach with the – by default – restricted types of data that can be collected from wild populations. The authors did exclude some data but these exclusions were based on the biology of the birds and justified statistically.

→ We are extremely grateful to the referee for their positive and constructive feedback.

I have several queries for the authors that may improve the paper from both the perspective of clarity but also strength of the results.

1) Much of the results and subsequent impact of the paper are shown in Fig. 2 and, in particular, panel F. I have several issues regarding this figure.

a. The legend provides N, eggs = 81297 and Nobs sperm = 1785. However, these are misleading because this value should represent the number of individuals measured. The finite number of individuals in the population means these values are repeated measures for females and females. I trust that all the R models incorporated individual as a random effect to control for this? Assuming that to be the case, the N values in the panel are overinflated and should be changed to reflect the number of individuals.

→ Thanks for highlighting that this may be misleading. We provide the number of individuals in Fig 1, but have also added this to Fig 2. We have adjusted the way means and standard errors are calculated in the figure such that it is based on individuals, and not individual-year combinations. This change is barely visible due to the high replication, but we agree with the reviewer that individual level is most appropriate. We originally included year-individual combinations because repeated observations of individuals are essential to addressing the questions in our manuscript. However, to avoid confusion we now report these in supplementary table 1 on a year by year basis.

→ All the statistical models include individual as a random effect as well as several other parameters (enclosure, year) to account for non-independence of data (see method section 3.4).

b. Panel F shows one of the most important results but I found it difficult to find in the methods the justification for the Y axis variable, “chicks/2 days”. How sensitive is this main result to variation in the Y axis variable? I would encourage the authors to be more clear in the justification for this in the main text and to refer in the main text to the area in the methods where this justification is statistically confirmed.

→ We agree this needs more explanation at an accessible position in the main text. Female ostriches can only lay an egg every other day. To correctly model the variance in this trait we therefore counted the number of eggs or chicks laid per two-day interval. This is necessary as binomial traits are expected to show reduced variance near 0% and 100%, and high variance at 50% success. If we used eggs or chicks per day, the highest probability in theory would be 0.5 instead of 1. We have added an explanation to the legend of figure 2 (line 113-115) and a reference to the section in the methods where this is explained, which we expanded compared to the previous version (line 361-364).

c. Related to panel F, I think it is very important for the authors to emphasize in the main text that incubation quality cannot explain this effect, since as it states only in the methods currently, that incubation is done artificially and therefore incubation quality is controlled for.

- Thank you for pointing this out. We completely agree and have added a sentence explaining this when hatching success is first mentioned in the results (line 133-135). We also considered including it in the Fig 2 caption, but the amount of information in the legend became overwhelming.

2) While I appreciate that the goal of the work was to assess individual reproductive success and its sensitivity to natural thermal fluctuations, the idea that cooperative breeding may offset individual consequences is interesting. While the authors state on line 253-254 the study was restricted to breeding pairs, it's unclear whether cooperative breeding is allowed on the site. If so, then there are several ways the authors can test the hypothesis that sociality buffers the population from individual reproductive consequences of thermal variation. Perhaps one way to do this would be to compare the outcome of episodes of cooperative breeding with paired breeding using the character state modelling approach (e.g., cold, benign, hot). Obviously because the authors understand the structure of their data better than me, I'm sure the authors can be clever in testing this hypothesis. While the work stands by itself without it, the impact of the work may be improved by formally testing whether flexible breeding systems could buffer climate change using data collected simultaneously on the reproductive outcome of paired and cooperative breeding attempts in a semi-wild population undergoing natural thermal variation. How cool! (or hot...)

- Very cool indeed! This is an important idea to test and we have a series of ongoing experiments to fully explore this. We are continuing to collect data from these experiments and so including analyses here would be premature.
- An important first step in establishing how social behaviour influences the ability of individuals to cope with climatic variation is to fully characterize the effects of temperature on males and females throughout the reproductive process, while controlling for social interactions. This is challenging under natural conditions, but we believe this study system and the analyses we present in this paper allow us to do this. We hope this will provide solid foundations for follow-up work investigating how other factors, such as social behaviour, both alleviate and exacerbate temperature effects on fitness.
- We have adjusted the wording in the caption of Fig 1 (line 77) and in the last paragraph of the introduction (line 66-68) to clarify that only one male and one female were present in each breeding enclosure. This is also described in the discussion (line 263).

3) I found it very interesting that there appears to be a mismatch between egg laying peak and thermal optimum for sperm number. The authors do not discuss the potential consequences of this. I find this surprising since the authors also state that male reproductive success in ostriches is contingent on high quantity and quality of ejaculates. Thus female reproductive optima is at a cooler temperature than male reproductive optima. Yet there are also females that specialize in reproduction at hotter temperatures. Do the authors have anything to say on this potential sexual conflict and female variation?

- This is an interesting point and one we wish we could explore more, but unfortunately, this dataset was not setup to address this question. Data were collected from pairs where male and female fitness is intertwined making it difficult to disentangle male and female effects. The differences in thermal optima are apparent, but the realised conflict between them depends on the width of the 'thermal neutral zone', which is extremely difficult to estimate accurately for each sex. Furthermore, the sperm data is collected from solitary males where there is no way of directly estimating how this influences female fitness.
- We have added a sentence to the results where we point to this interesting contrast (line 94-96) and that it requires more targeted follow-up investigations. We also explain in the methods why the dataset is not setup to examine sex differences in thermal optima (line 402-405).

4) Minor comments:

a. Line 243 has a referencing issue. Gilchrist...

- Addressed.

b. Line 303 requires a reference to justify the statement that male reproductive success in ostriches to deliver high quantity and quality ejaculates.

- We agree this is needed. This is a general phenomenon in birds and so we have adjusted the wording of the sentence to make a broad statement about birds, rather than specifically ostriches, and supported this with references (line 319-320). We do have data to also support this in ostriches, but it is currently unpublished.

c. Line 347 appears to have a typo: with without

- Thanks and done.

Reviewer #2 (Remarks to the Author):

This is a really interesting study, with a great dataset, investigating important questions about how reproductive traits are impacted by changes in temperature and how much this varies across individuals. I think it will be of broad interest to those interested in both evolutionary and climate change biology. The methods appear to be thorough and robust, and the conclusions drawn are mostly fair and well considered. The paper is nicely written and easy to follow, and does a good job of putting these results into the wider context of how fluctuating temperatures may impact bird reproductive success as the global climate continues to change. I do have a few questions for the authors and some suggestions for improvement, but I think these can mostly be addressed by adding some more information/explanation to the text and making some relatively minor amendments. I'll first mention two broad points and then go through more specific comments on a line-by-line basis.

→ Thank you very much! We really appreciate the thorough and positive feedback.

Broad comments

My first broad comment is that I am not convinced of the premise, laid out in the introduction, that the effects of temperature change on reproductive success is little studied in birds – in fact, there are a number of studies that have addressed this question in different ways (e.g. Järvinen 1994 *Ecography*; Kitaysky & Golubova 2001 *J. Anim. Ecol.*; Julliard et al. 2004 *PRSB*; Weatherhead 2004 *Global Change Biology*; Auer & Martin 2012 *Global Change Biology* – list is not exhaustive), as well as a wealth of poultry science studies showing such effects experimentally. This doesn't detract from the current study, but rather places it within a growing and important topic that is of broad interest.

→ Our intention was not to give the impression that temperature effects on reproduction in birds has not been studied. Rather to highlight that previous research on birds has primarily focused on temperate species and reproductive phenology. We appreciate the referee for bringing this to our attention and have now heavily revised this paragraph (line 50-60). We more clearly emphasize that previous research has been on temperate species, citing the references indicated by the referee.

My second broad comment is that I think it should be recognised, in the discussion and perhaps in the abstract/intro too, that ostriches are a special type of bird, and several ecological and physiological factors important for them may not be applicable across birds more broadly. For example, ostriches are already well adapted for breeding in hot arid conditions (they could be considered already relatively 'heat-stress adapted'), and indeed several characteristics of their eggs help them cope with such conditions. Ostriches also have relatively small eggs for their body size and therefore relatively low egg investment, which may mean they are less likely to be subject to intense trade-offs between egg size and number, even under stress. The paper attempts to make some broad conclusions about thermal stress representing a "considerable and underappreciated factor limiting avian reproductive success", but I would like to see these conclusions tempered slightly to acknowledge that these issues could (and may be likely to) be quite species-specific. Again, rather than detracting from the study, this point actually highlights the importance of species-specific studies like this.

→ We agree that ostriches are a special type of bird and that this is important when drawing more general conclusions. As requested, we have tried to be more careful with our conclusions about whether our results can be generalized across species (line 269-270 & 272-274). We raise this as an open question that we hope will stimulate further work and caution about ostriches being 'typical' given their unique biology.

→ The small egg size relative to body size is a good point and we have now included a statement about this when discussing trade-offs between egg mass and number (line 254-258). It is worth noting that although egg mass is relatively small, ostriches naturally lay a large number of eggs per year compared to many other species (>80 eggs per year). Predictions about ostriches being less likely to experience trade-offs are therefore not straight forward.

Specific comments

line 19: "(-5°C - 45°C)" this should perhaps be changed to something like "below X°C and about Y°C" since you are referring to extremes but this range includes 'normal' temps

→ Agreed, we have adjusted the wording to "ranging from -5°C to 45°C".

line 20: I don't see that there's any immediate reason to expect an effect of elevated ambient temperature on gamete quality, since all bird sperm are produced inside the body at high temps

- The internal body temperature of ostriches is 39.3°C (Maloney, 2008 Australian Journal of Experimental Agriculture) while external temperature exceeds 52°C at times. This creates the potential for temperatures to directly and adversely influence sperm viability via thermal stress, dehydration and other physiological mechanisms that make it difficult to maintain homeostasis during periods of elevated heat stress. Even when ambient temperatures do not exceed internal body temperatures, heat produced through metabolic processes, together with an inability to dissipate heat, can cause physiological imbalances that may impede sperm function (Chickens: Karaca et al. 2002, British poultry science; McDaniel et al. 1995, Poultry Science). However, our intention was not to insinuate that ambient temperature would directly affect individuals' gametic cells, but rather that temperature effects would be mediated through general physiological changes. For example, the physiological costs of protecting essential somatic functions at high temperatures can cause a general down regulation of reproductive function that may impact sperm viability (Sgró & Hoffmann 2004, Heredity; Walsh et al. 2019, TREE).

line 34: it is important to distinguish here between fertility specifically (i.e. effects on a focal individual's ability to produce functional gametes – which is what you appear to be testing) and prenatal death of a zygote/offspring. Some of the papers referred here don't actually differentiate the effects of temperature on traits that affect fertilisation success/failure from those that affect surviving offspring produced ("fertility" can be used as a catch-all term). Failure to distinguish these two things will mask the mechanistic basis of any temperature-mediated impacts on reproductive success. This distinction will help to inform several of your other hypotheses too.

- We agree this distinction is important. We have a measure of sperm viability, but also of zygote viability in terms of hatching success. We have adjusted the wording in the introduction to clarify that temperature can influence both (line 34).

line 55: a recent study by Hällfors et al (2020, PNAS) showed that breeding periods are often not just advancing but also contracting – this may have important implications for reproductive outputs, especially for species that produce >1 clutch or frequently fail on first attempt and re-lay

- This is an important finding and we now cite this paper in that paragraph (line 51). Note that this paragraph has been heavily revised due a request from another reviewer.

line 69: this is a really great dataset

- Thanks!

line 74: reproduce repeatedly across the year in response to what? is onset of breeding directly/indirectly driven by temperature?

- Although ostriches are known to breed all year round, previous studies on environmental correlates of reproduction in the wild have proved inconclusive (Jarvis 1985 Ibis; Magige 2009). That said, most breeding attempts occur in the dry period before seasonal rains, potentially to time hatching with increases in vegetation for feeding chicks. However, what cues are used to initiate reproduction during the dry season is unclear.

line 76: are all the birds in this study captive? i.e. all 1299 birds studied (line 71). It would be useful to clarify if this is a wild, semi-wild/managed (if so, how?), or captive population in the main text, to help the reader fully understand the system set up

- They are captive and each breeding pair is kept in an enclosure of semi-natural habitat. We have now clarified this in the main text (line 67-68) and in the methods (line 286).

line 77: are the solitary males also in enclosures? Are they kept alone? Being alone and isolated from other individuals (both male for competition and females for sexual arousal) may influence sperm production.

- Solitary males are kept in enclosures in visual and acoustic contact with other males and females. Previous work on isolated males and females shows that sperm production and egg laying rates vary seasonally, which closely matches the reproductive success of breeding pairs. We therefore believe that the data collected from these males is not adversely influenced by them being physically isolated. Isolating males is also essential as it allowed us to control for copulation history and effects of sperm depletion (see response to next comment). We have clarified this in the main text (line 68) and have added details on how males were kept to the method section (line 291 & line 320-324).

line 95: sperm number is likely to be influenced by how frequently males have recently ejaculated – presumably

this will differ between paired and unpaired males (and perhaps between different paired males too, depending on how consistent copulation rates are across pairs)? Lack of ejaculation could also influence sperm viability and this could be a particular problem for solitary males – fewer ejaculates = older deteriorating sperm? Was sperm collected from solitary males regularly to account for this? This is not clear from the methods section either.

- Isolating males allowed us to carefully control the copulation history to avoid any effects of sperm depletion or prolonged periods of sexual rest. Data on ejaculates used in analyses were typically from continued monitoring of males, where ejaculates were collected 3-5 times per week. This is within the range of copulation rates observed in pairs (pers obs). On some occasions other experimental work was being performed where multiple ejaculates were collected from individual males per day. As described in the methods, on such occasions we only included the first collection. We have now provided more details in the methods section 2.2. about the copulation history of males used for sperm collection (line 320-324).

line 109: although egg mass was unaffected, other egg traits e.g. egg composition may have varied, without necessarily affecting overall mass/size. This can also be important for hatching success. A brief point on this would be useful in the discussion.

- Agreed and thank you for pointing this out. We have added a point about this in the discussion (line 236-237).

line 111: I think this is no surprise really, so perhaps not "remarkable" – as mentioned earlier, in birds the testes are inside the body, so sperm are made in very warm conditions! I don't think there is much argument here for sperm quality being directly impacted by ambient temperature. Perhaps for ambient temperature affecting other aspects of the male's physiology, with knock-on effects for sperm production (but in that case, we might expect it to be numbers rather than sperm morphology). I'd like to see this point clarified in the discussion. Linked to this: would we necessarily expect the impact of higher vs lower temps on gamete production to be the same? Mechanistically, surely not.

- We have removed the word "remarkable" as requested.
- For justification for examining the effects of ambient temperature on sperm viability given high internal body temperatures see line 40-46 & line 218-219 where we now cite Karaca et al. 2002, and our response to the comment above ("*line 20: I don't see (...)*"). We would like to emphasize that although birds are endotherms and considered to have a stable internal body temperature irrespective of ambient temperature, maintaining such homeostasis in the face of extreme environmental temperatures, such as during heat waves, is costly and can result in stress that is likely to affect sperm viability and other physiological traits.
- As the reviewer points out, corresponding declines in sperm number and viability with temperature stress would potentially indicate a general impact of temperature stress on male physiology, which is clearly not the case. Further work is nevertheless needed to establish if the different patterns in sperm numbers and viability are due to these traits having different sensitivities to male physiological stress or due to different mechanistic control. For example, the time taken for sperm cells to develop is thought to be largely invariant and the blood testes barrier can protect sperm cells from physiological stress (Mita et al. 2011). In contrast, the regulation of sperm production is clearly influenced by external environmental factors. Similarly, whether heat and cold stress have similar impacts on sperm numbers depends on the mechanism, but the fact that we see similar responses to hot and cold temperatures suggests that declines in sperm production are linked to physiological costs associated with thermoregulation. We have now extended our discussion of the mechanistic insight provided by our results along these lines (line 226-240).

Fig 2: the deviation range for decreasing temps seems much narrower than the range for increasing temps. Why is the shift from the optimum not consistent across plots (are different data used for each?). Looking at your methods, it appears that this is probably explained by sperm being sampled over a more limited (but still impressive!) period of time - if this is correct, please spell out the sample sizes for the different analyses more clearly.

- Yes, the difference in range is due to the more limited sampling of sperm. We have added a sentence to the caption clarifying this (115-116), and provided a supplementary table with yearly number of individuals sampled for each trait (Table S1).

Fig 2: Is (F) needed? If hatching success is not strongly influenced by temp, then doesn't the decline in offspring (F) simply reflect the decline in egg numbers (A)?

- We agree that when there is little effect of temperature on hatching (C), then the null-expectation is that temperature effects on numbers of offspring (F) will equal those in egg numbers (A), as the referee states. However, there is substantial variation in hatching success among females, and if this is in any way non-random with respect to egg-laying rates at different temperatures then this may alter patterns of offspring number. For this reason, and because offspring number is a more widely used proxy of fitness, we think it is an important to keep this panel.

line 145: I'm not sure it is valid to conclude "cumulative detrimental effects on eggs and sperm" - yes for eggs, but why sperm? Egg numbers decline and this drives the decline in offspring number. It seems to me that the effect on hatching success is limited and since developing embryos are highly sensitive to temperature and associated humidity changes, any impact on hatching success could be direct effect on embryo survival (i.e. nothing to do with fertility or sperm). Developing eggs are particularly sensitive to humidity fluctuations (even during artificial incubation, since many artificial incubators do not stand up well to extreme ambient conditions when auto-regulating temp/humidity).

- We agree that there is potential for other factors to influence our results. However, there are significant reductions in sperm number and hatching success. We therefore think it is important to convey that the effects of temperature on egg and sperm numbers will combine to cause greater declines in reproductive success than if we consider egg numbers alone.
- The artificial incubators are capable of holding both humidity and temperature constant, but we agree with the referee that external humidity can still affect incubation success. However, this will influence average rates of hatching rather than influence our results. We are examining temperature conditions 0-4 days prior to laying and relating this to hatching success. After egg collection, eggs were stored for 1 to 7 days and then incubated for 42 days. The humidity conditions the egg experiences are therefore decoupled from our main predictor – temperature prior to laying.
- Given these facts, we believe there is enough evidence to support our interpretation but have added the word "suggest" to make our interpretation more cautious (line 144-146):
 - *These results suggest that the negative effects of temperature fluctuations on reproductive success arise through the cumulative, detrimental effects on eggs and sperm under both low and high temperatures.*

line 149: these are such interesting results – it's as though there may be 'subpopulations' of individuals with slightly different trait profiles. Would be fascinating to know what the mechanistic drivers are (I hope to see this in a future study!)

- Thanks! We hope to dig deeper into this as well.

line 162: I wonder if this is because egg size is relatively small in ostriches (for their body size), so there is not as much constraint and it is relatively easy to maintain egg size even under stress? Maybe you would see more dramatic effects in a bird with large relative egg size, as this egg requires relatively more investment by the female?

- This is a very good point. We have now included this in the discussion of differences in response among traits (line 254-257).

line 168: I think egg size (and shape) is probably mostly determined (and constrained) by the size/morphology of the female reproductive tract, so this makes good sense to me

- Agreed, this may be the constraint.

line 174: are these fertility related traits always depressed at the individual level when temperature deviates from the optimum, or do some individuals specialise to the extent that they 'improve' with slight increases/decreases in temp? This is what your phrase "some females appeared to specialise at reproducing at high temperatures" (in abstract, line 21) implies, and if so it's very interesting indeed and I would like to see the point be made more clearly.

- Yes, there are several individuals that increase laying rates when temperature increases beyond the optimum. With the phrase in the abstract we refer to two pieces of evidence: 1) there is a lot of variation in the thermal plasticity among females, so some will lay many more eggs at high temperatures than others, and 2) females that have relatively high laying rates at high temperatures also lay larger eggs. While this evidence shows that some females produce more and larger eggs at high temperatures than others (now clarified in line 195-197), another reviewer requested that we toned down the use of "specialization" in the abstract. We have tried to strike a balance here by describing the results as clearly as possible and removing any more loaded terms such as specialization.

line 192: this is another example where what we see in ostriches may not be so easily applied to other birds. Temperature stress may typically play a very important role in inducing trade-offs between egg numbers and egg mass in a species where producing large eggs is relatively hard work/high investment. As mentioned above, ostriches have relatively small eggs for their body size, so relatively low investment, and this may mean it's just not such a big deal for them.

- We are once again grateful to the referee for raising this important point. While we agree that the relatively small size of individual eggs is important to consider here, we believe the picture may be more nuanced (see also the response to comment above starting with "My second brood"). Whether trade-offs between different egg traits occur (e.g. size and number) will depend on whether females are limited in their total reproductive investment and how reproductive costs accumulate across offspring, rather than being a simple product of investment per offspring. For example, if species produce large numbers of offspring then small costs per offspring can accumulate to significant levels, especially if the total resources individuals have to invest in reproduction is limited by living in extreme environments. For these reasons, we think trade-offs between egg number and mass may be just as pertinent to ostriches as to other species.

line 205: was it six adult individuals that died from heat stress? What about embryos/chicks/juveniles (can you tell)?

- Yes, six adults. We have changed the word "individuals" to "adults" (line 208). The effects of heat stress on embryos is controlled for in our dataset through collecting eggs twice a day and by artificially incubating eggs. We do have data on chick survival, but unfortunately it is not possible to tell if heat is the cause of death.

line 208: cooler temps are likely to be more of a problem for species already adapted to hot environments (like ostriches). I also (again) wonder about what triggers the onset of reproduction in this species, given that they breed opportunistically throughout the year. Is it known why they choose certain times to breed? Temp may be an important mechanistic trigger (proximate cause).

- We agree that cooler temperatures may be stressful for species adapted to the hot temperatures and that we know little about this. This is an important aspect of our work, which supports the idea that decreases in temperature are just as important as increases in temperature for determining reproductive success (Fig. 2). Given that previous work has focused on temperate species and the effects of high temperatures, it is difficult to say whether this is more or less important for species living in hot, arid environments. It is worth noting that in such environments cold temperatures, particularly at night, are frequent (Fig. 1).
- With respect to what triggers reproduction in ostriches, little is known beyond breeding being more frequently in the dry season (see above comment starting with "line 74: reproduce"). Which triggers are important for the initiation of reproduction and how they elicit reproductive behaviour is an important future avenue of research.

line 221: I think you should clarify that there is no sign of an upper thermal limit *within the natural range of temperatures that the birds encountered during the study* (there will, of course, be a limit)

- Agreed, this was unclear, we have removed the term "upper thermal limit" as in some fields (e.g. ectotherms) this is the point of zero survival/performance, which is inappropriate for a trait such as egg mass. See line 223-224).

line 222: how was egg number was adjusted? Did females lay smaller clutches, or fewer clutches, or stop laying sooner, or widen gaps between eggs... again, wondering about the mechanisms that underpin these effects

- This a great question that we hope to dive into in the future. Preliminary analyses suggest that both gaps between eggs within clutches and the decision to initiate as well as terminate clutches is adjusted. We did not look into clutch size, but as they are a product of the former parameters, we find it likely that they are reduced.

line 228: again, it's also worth noting that these gamete production processes already take place at much higher temperatures in birds than most other animals, because body temp is already high (~40°C in passerines – I don't know if that is true for ostriches!) and testes internal in birds. Sperm production in birds *must* already be adapted to high temps relative to most other species. All the references listed here are mammalian studies, were testes are relatively external and therefore cooler.

- The distinction between birds and mammals is clearly important. We now point out that this knowledge is from mammalian studies as we could not find similar experiments in birds (line 228). We have also added a sentence stating that inherent reductions in heat sensitivity may also explain the small effects on sperm viability (233-234). See also our response to the comment above (starting “line 20: I don’t”) with respect to high temperature still being relevant to sperm production in birds.

line 245-246: this answers my earlier question about whether some females actually did better in elevated temps: it’s such an interesting finding, but I didn’t feel it was entirely clear from your results (to me at least). Can you make this clearer (is it possible to visualise in some way)?

- The reviewer is referring to the original sentence: “*In fact, the opposite was true, with certain females increasing reproductive investment in egg laying during hot periods, with no apparent reductions in fertility at other times.*” With this sentence we are referring to the positive phenotypic correlation between egg-laying rate and egg mass (“investment”) as temperatures increase (Fig 3), and that no trade-offs were found among any traits at any temperature. This shows that the individuals that were best at maintaining egg-laying rates at high temperatures also laid heavier eggs at higher temperatures. We have clarified this section in the results (last sentence of the results, line 195-197). As stated above some individuals do lay more eggs at high temperatures than at benign, but another reviewer asked us to tone down the focus on specialization (see comment above starting with “line 174: are”).

line 261: it is crucial to quantify the effects of temp on fertility of species in different biogeographical zones, but I would also argue that we must consider how this may differ across different reproductive strategies e.g. it might not be too bad if you are an opportunistic breeder and can wait for conditions to be 'right' but if you are a seasonal breeder, suboptimal temperatures during the breeding season could be extra costly. In addition, as said before, in species where there may already be a stronger trade-off between reproductive traits e.g. where egg size is relatively large and therefore a bigger investment, the impacts of temp might extend to gamete quality as well as quality.

- We completely agree and have included “breeding biology” to the sentence marked by the reviewer (line 272).

line 269: am I inferring correctly from this that the ostriches are research/farmed animals, not wild? It is important to make this clear both here and at the relevant point in the main text. The fact they have ad lib food, water, etc etc as a result of being captive has the potential to make a big difference and is something that should be addressed in the discussion (the impacts of temperature on reproductive success could be far more dramatic in wild systems)

- This is another good point. The population is used for research and are neither domesticated nor free-roaming animals. We have added the word “captive” to the relevant method section (line 286) and to the last paragraph of the intro (line 67). We have also added a part to the results to highlight that the effects we detected could be more dramatic in free-roaming ostriches where food and water are likely to be more limited (line 146-147).

line 276: please provide sample size of males from which sperm samples were analysed here, and above please provide sample size of breeding pairs monitored (which presumably = number of females analysed for egg production?)

- Done (line 291 and 286). Females received a new partner when possible (line 289-290) and number of unique pairs is therefore not equal number of unique females across the two decades. As number of females are the relevant unit of replication in the analyses, we provide this number.

line 300: please provide final sample size for females analysed after all this data removal

- Done (line 316)

line 305: were the males routinely ejaculated at regular intervals on the run up to collecting the experimental samples? Wondering about how 'fresh' the sample was (males can store old sperm for some time if they don't have the opportunity to ejaculate regularly)

- See response to comment above starting with “line 95: sperm”.

line 311: I'm not sure this is convincing with respect to overall 'viability' – sperm that appear morphologically normal are often non-motile, sperm with non-intact membranes are sometimes motile; these assays are not a perfect measure of sperm viability and fertilising ability. I would prefer for the specific sperm traits to be referred

to both here and in the main text, to avoid misleading the reader i.e. temperature didn't affect sperm morphological normality or sperm membrane integrity. Until I got to the methods, I was assuming some sperm motility analysis or similar had been performed.

- Agreed the term “viability” can be misinterpreted. We have added a definition of sperm viability “(*viable sperm: normal morphology, intact membrane and eosin impermeable*)” to the first mention in results to avoid misleading the reader in the results (line 104-105).

line 315: analysing failed eggs for fertilisation success would have been a better measure of sperm function/success, because it is the only way of demonstrating that the sperm actually made it to the egg. This is why when you are talking about hatching failure in the main text you cannot conflate it with “infertility” – it could be embryo mortality

- We agree that analyzing failed eggs for fertilization success would have provided better data on sperm function. However, this was not practically possible for such a large number of eggs as it is difficult to distinguish between early embryo mortality and infertility without careful training of staff. There are also many other reasons for infertility other than poor sperm function (e.g. mating incompatibility is prominent in this species). Consequently, we have tried to be careful in not using the word “infertility” in relation to our measure of “hatching success” in the MS. When presenting the results from hatching success in the results we explain that this measure is a product of both male fertilizing ability and egg viability (line 131-133).

line 333: I agree that thermal windows after egg laying should not affect sperm or egg production, but could they not influence hatching success e.g. if elevated (or reduced) temperatures are more likely to result in early embryo death (which lots of poultry science literature suggests they are)? See Christensen 2001 World's Poultry Science Journal for a review.

- Agreed, they definitely could. To reduce this possibility, eggs were collected every 6 hours during the day and incubation conditions were carefully controlled using an on-site hatchery (line 302 & 336-342). Having such control over conditions after laying allows pre- and post-post laying processes to be separated, which is a real strength of this dataset. This info has now been added to the main text at the first mention of hatching success in the results (lines 133-135).

lines 326-355: is anything known about what is happening physiologically (i.e. inside the bird) with respect to gamete production during these critical periods that have been identified using this method? It strikes me that it might be important to consider what the critical phases in gamete production might be, and that this should be fed into this somehow.

- Yes, as mentioned in line 366 the two days before egg-laying is when the egg travels down the oviduct. However, more in-depth studies are needed in the future to fully understand the physiological links between heat stress and reproduction.

Title: I would prefer “fertility” to be replaced with “reproductive traits” since this is more accurate (the study doesn't explicitly measure fertility in terms of the ability of males/females to produce fertilised eggs).

- We are grateful for this suggestion and have carefully considered it. The work presented in Walsh et al TREE “The impact of climate change on fertility” is a major inspiration for our study. They use the term fertility to capture the different ways viability of offspring can be affected (e.g. gamete viability and gamete number, see their Table 1 column 4+5), many of which we investigate here. We feel that “reproductive traits” includes a wider range of traits (e.g. morphological characteristics of reproductive organs, sexually selected ornaments etc.) and therefore comes with a higher risk of misleading the reader.

Nicola Hemmings

Reviewer #3 (Remarks to the Author):

Using two decades of data, the authors investigate whether extreme temperatures have damaging effects on different fertility traits, explore individual variation in resilience of fertility to changing temperatures, and test whether extreme temperatures cause trade-offs in investment across traits. The novelty of this study comes from the fact that the authors are using a powerful dataset to test how temperature fluctuations affect fertility traits, rather than survival, and working in a tropical/sub-tropical system, rather than a temperate system. Interestingly, the authors find that only some fertility traits are sensitive to temperature extremes, and that some females appear to specialise in reproducing at higher temperatures, which could facilitate responses to climate change - a finding that will certainly influence thinking in the field. Overall, the manuscript is very well written, the statistical analyses appear to be robust, and the topic will be of broad interest to readers.

→ We thank the reviewer for the nice words and constructive feedback!

I only have a few minor comments:

1. How were the birds paired each year? Was it random? How often were the same individuals paired up over years? Or was this avoided?

→ Birds were paired to avoid inbreeding but otherwise pairing was random. On average 25% of pairings each year were between individuals that had previously not bred together. However, as the same females were monitored for several years, more than half were paired with at least two different partners. We have described this strategy in the methods (line 289-290).

2. What was the degree of relatedness? Were offspring from some years put into the breeding programme? If so, is it possible to look at whether offspring of resilient females are also resilient?

→ The individuals are derived from 139 founding individuals (line 283) and so the population does contain related individuals. We are currently working on pedigree analyses for the population, which we hope to pursue in the future.

2. I could not really follow the sample sizes. There were 1299 individually marked birds across the 20 years, and data was collected from 650+22 (Figure 1B) of them? Further, I do not completely follow the n individuals by year = 80 in Fig 1 B, but then n = 18, 49 individuals by year combinations in the figure legend. I'd like to see a better breakdown of the sample sizes within and between years. This would allow to reader to understand how long individuals were followed and what the age structure was.

→ We thank the reviewer for highlighting that greater clarity is needed in reporting sample sizes, which is extremely important to us. The 1299 individuals referred to in line 64, refers to total number of birds used in the study (number of males and females in breeding pairs and the number of solitary males). In figure 1 we aim to give a quick and simple overview of the individual replication and repeated sampling across years and across traits. We have changed the way we provide these numbers, such that we now give 1) number of females/males monitored for egg/sperm data; 2) number of years that egg/sperm data was collected and 2) average number of years a female/male was monitored.

→ We have also added a new supplementary table 1 that gives an overview of the number of individuals sampled in each year for each trait.

→ The different numbers in the figure legend reflects that not all males assessed for sperm numbers were also assessed for sperm viability. We have now adjusted this sentence to avoid any misunderstanding (line 79-81).

Reviewer #4 (Remarks to the Author):

This MS uses a very impressive dataset to evaluate the impact of temperature on fertility in ostriches, under semi-natural conditions. It finds that both male and female fertility are substantially impaired under naturally occurring temperature variation, by both cold and hot temperatures. I am unaware of any similar studies of this magnitude, and see this MS as a major advance in the field.

Overall, I am strongly in favour of publishing this MS. I think the dataset is compelling, and the conclusions reasonable and interesting. I have a two issues that I think need to be resolved, plus several other points that are more optional but that I think will improve the MS.

→ We are very grateful for the supportive and constructive comments. Many thanks!

Major issues (although I think they should be reasonably easy to resolve):

L96-98 and methods: there is no justification given for why a 3 day window is appropriate. Is there literature suggesting this is a good idea? Did you try other window sizes? I think you need to either show this doesn't matter, or provide evidence/literature that bird fertility is affected by windows like these. Quail spermatogenesis takes 13 days according to Lin & Jones (1992 doi:10.1007/BF00319382), and I can't see any reason to assume mature or nearly mature sperm should be particularly vulnerable to thermal stress, so limiting your range of potential windows to 7 days before mating also seems odd. It is entirely possible that mature sperm could be robust and early stage sperm are especially vulnerable. As far as I know, your work is the first to establish a time point for when thermal stress impacts on bird sperm- so in my view it is important to do it right. If you get this wrong, your paper will be used to justify a 3 day window by any other bird people interested in similar questions, which will cause problems for many other researchers.

- Window size: We are also unaware of any other studies that have tested the influence of thermal time lag effects on reproduction in birds. Our analyses were therefore designed to objectively quantify the time sensitivity of egg and sperm traits to thermal fluctuations without prior expectation. The challenge here is that choosing large windows starts to capture seasonal trends whereas choosing very small windows may miss extreme temperature events that influence reproductive success. We systematically investigated different window sizes (1 to 3 days) to examine the best analytical approach to capture time lag effects. This was done by using generalized linear models and examining AIC values from models. These analyses showed that the temperature of days 4, 3 and 2 before laying were important, and that a three-day window was best at capturing the short-term thermal fluctuations in these days (**Fig S1A and S7**), while a larger window size dilutes the signal. We have now included this new figure in the supp material (**Fig S7**) and refer to it in the methods section when presenting the three-day interval approach (line 354-356). We have also added that the 3 day window is specific to this study and is something that should be estimated for other species rather than used as a 'rule' (line 387-289).
- Days prior to sperm collection: This is an important issue and we are grateful to the referee for helping us clarify and investigate it further. We have now expanded our analyses to include up to 15 days prior to sperm collection instead of the previous 7 day limit. This confirms our previous findings, which we now report in the methods section (line 379-381, **Fig S2**).

I think your argument for thermal specialisation in females is fairly weakly supported by your data. I think you should tone down this argument (but still mention it- it is a very interesting idea). I think your MS is worth publishing even without this idea, so I would include it as an interesting possibility consistent with the data rather than the stronger version you give. Alternatively, convince me (and the other reviewer and editor) that you are right.

- The conclusion that some females specialize in reproducing at high temperatures was based on two pieces of evidence: 1) some females repeatably lay more eggs at high temperatures than others, and 2) females that lay more eggs at high temperatures also lay larger eggs (now clarified in line 195-197). The statistical support for these results is robust, but we realize that "specialization" is a loaded term that, as the referee points out, may require more in-depth investigation. We have toned down the wording in the abstract from "specialize" to "invest more", illustrating the result instead of extrapolating to interpretation (line 23).

Minor issues:

L54 You are overstating this- there is a long literature examining the direct effect of temperature on sperm in mice, flies and livestock, for example. In general this paragraph is poor compared to most of your writing. It mashes ecological/phenology studies on birds (largely about food supply as you mention later) with temperate vs tropical. I suggest you rewrite this. 1) make a clear "most studies are on temperate species (or even specify

temperate birds), so we need more information about tropical species potentially adapted to generally high temperatures". 2) A lot of work on birds has focused on adjustment of breeding time, which may be mediated by food availability, but there is also a need for more direct studies on sperm and egg traits which can also be directly affected by temperature at ecologically relevant temperatures (Hurley et al 2018). This is what you actually do, and it is worth contrasting how different this is to phenology.

- We thank the reviewer for this suggestion. We have now re-written the entire paragraph along the lines suggested by the referee (line 50-60)

L70 "in a tropical species, the world's largest bird, the ostrich (*Struthio camelus*)." makes it clearer you don't mean "the world's largest bird that lives in tropical countries" ie there are bigger birds in temperate zones.

- Addressed as suggested.

L90 probably worth changing this to "Geographic range (green) of the" to make it as easy as possible for readers.

- Addressed as suggested.

L91 "mean monthly range" could mean a wide variety of things (mean of the range of temperatures experienced in a month/mean of the max temperatures experienced in a month/mean difference between mean monthly temperatures in a year)- better to be really specific about what you mean.

- Thanks for highlighting that this is unclear. The number we are presenting is: mean(range(July), range(June), range(May) ...). We have changed the wording to "Monthly range was calculated by estimating the range of temperatures of each month and then calculating the mean of this across all months" (line 82-83).

Fig 2 I don't understand why you have ordered your graphs (A-F) completely differently to how you explain them in the figure legend (A, B, D, E, C, F).

- We have reordered the graphs to match the order in caption (and text).

L154-155 Wow!

- Agreed!

Fig 3 Also wow!

- Agreed!

L243 omit George W

- Done

Discussion- I think you need to briefly discuss whether ostriches are a good model for other organisms, or are weird. Do you expect similar effects in non-ratite birds? How about mammals as they are also endotherms?

- As detailed in the introduction we are not using ostriches as they are a good model for all other birds (or endotherms), but as we know almost nothing about if, and how, birds in tropical or sub-tropical regions are able to reproduce under the highly fluctuating and extreme temperatures they are exposed to. Given the little comparable information on other birds and mammals, it remains an important empirical question as to how the reproductive success of other tropical and sub-tropical species varies with temperature. We hope our study will help stimulate this type of research.
- We have clarified in the discussion which of our results are consistent with general theory, and therefore potentially generalisable across species, and which results are more likely to be dependent on the unique biology of ostriches (lines 268-279, see also line 254-258).

L300 this would be stronger if you gave the mean number of eggs, or the number of pairs excluded vs included, to show this was a reasonable thing to do.

- We have added the number of females in the dataset before (line 286) and after filtering (line 316).

L340 it would be interesting to know if you think the difference in thermal windows in egg viability/size and male sperm number/viability indicated different biological processes and genuinely different vulnerable points, or if you

think the viability and hatching data are unreliable because the effects are so much weaker

- ➔ We investigated the robustness of the identified thermal windows for each trait to see if they could be compared in a meaningful way. In summary, we found that estimates of egg mass and sperm viability were quite sensitive to outliers and which covariates were included in models (see new methods lines 367-375). This was not the case for egg numbers and sperm numbers. We are therefore confident that thermal windows of egg numbers and sperm numbers reflect interesting biological processes, but refrain from drawing comparisons with the windows of egg mass and sperm viability as these are less reliable.

Tom Price

Reviewers' Comments:

Reviewer #1:

Remarks to the Author:

The authors have addressed all my outstanding comments adequately and, to my mind, also the other reviewers. I thank the authors for their care in the response. The paper represents very interesting and well-designed research and I enjoyed reading it.

Reviewer #2:

Remarks to the Author:

The authors have done a great job of addressing my comments/answering my questions, and those of the other reviewers. I think the changes they have added significant clarity to the manuscript. I have no further comments. I congratulate the authors on a very impressive and interesting study, and look forward to seeing it in print!

Reviewer #3:

Remarks to the Author:

The authors have done an excellent job at addressing reviewer comments, and the manuscript reads well.

Reviewer #4:

Remarks to the Author:

I am happy with the changes they've made, and have no further comments to add. It's a really interesting paper, and I suspect the follow on papers once they have the pedigree will be even better.

Tom Price